# ReConcile: Round-Table Conference Improves Reasoning via Consensus among Diverse LLMs

## Abstract

Large Language Models (LLMs) still struggle with complex reasoning tasks. Motivated by the *society of minds* (Minsky, 1988), we propose ReConcile, a multi-model multi-agent framework designed as a round table conference among diverse LLM agents to foster diverse thoughts and discussion for improved consensus. ReConcile enhances the reasoning capabilities of LLMs by holding multiple rounds of discussion, learning to convince other agents to improve their answers, and employing a confidence-weighted voting mechanism. In each round, ReConcile initiates discussion between agents via a 'discussion prompt' that consists of (a) grouped answers and explanations generated by each agent in the previous round, (b) their uncertainties, and (c) demonstrations of answer-rectifying human explanations, used for convincing other agents. This discussion prompt enables each agent to revise their responses in light of insights from other agents. Once a consensus is reached and the discussion ends, ReConcile determines the final team answer by leveraging the confidence of each agent in a weighted voting scheme. We implement ReConcile with ChatGPT, Bard, and Claude2 as the three agents. Experimental results on various benchmarks demonstrate that ReConcile significantly improves the reasoning performance of the agents (both individually and as a team), surpassing prior single-agent and multi-agent baselines by 7.7% and also outperforming GPT-4 on some of these datasets. We also experiment with GPT-4 itself as one of the agents in ReConcile and demonstrate that its initial performance also improves by absolute 10.0% through discussion and feedback from other agents. Finally, we analyze the accuracy after every round and observe that ReConcile achieves better and faster consensus between agents, compared to a multi-agent debate baseline.[1]

## 1 Introduction

A large body of work has focused on improving the reasoning capabilities of Large Language Models (LLMs) by imitating various human cognitive processes. These include phenomena like reflecting on and critiquing one's own predictions, being receptive to feedback, and learning from feedback. Of note, self-reflection is an introspective process that allows the model to improve its outputs by generating feedback from the model itself (Madaan et al., 2023; Shinn et al., 2023). However, self-reflection suffers from Degeneration-of-Thought – when the model is overly confident in its answer, it is unable to generate novel thoughts even after multiple rounds of feedback (Liang et al., 2023). Moreover, it is difficult for the model to refine knowledge that it already does not contain.

To promote more diverse thoughts, past work has drawn inspiration from the *society of minds* in multi-agent systems (Minsky, 1988; Zhuge et al., 2023). Communication between multiple agents plays a vital role in complex decision-making. This has prompted recent developments of multi-agent debating frameworks (Liang et al., 2023; Du et al., 2023), in which multiple agents participate in a multi-round debate to arrive at a common final answer. Despite the increased reasoning diversity obtained through the process of a debate, multiple agents have typically been limited to different instances of the same underlying model, ChatGPT (OpenAI, 2022).[2] This

---

[1]Code for all experiments is provided in the supplementary.

[2]In this work, we refer to multi-agent as multiple instances of the same underlying model (e.g., ChatGPT), whereas multi-model model-agent refers to different models (e.g., ChatGPT, Bard and Claude2) as agents.

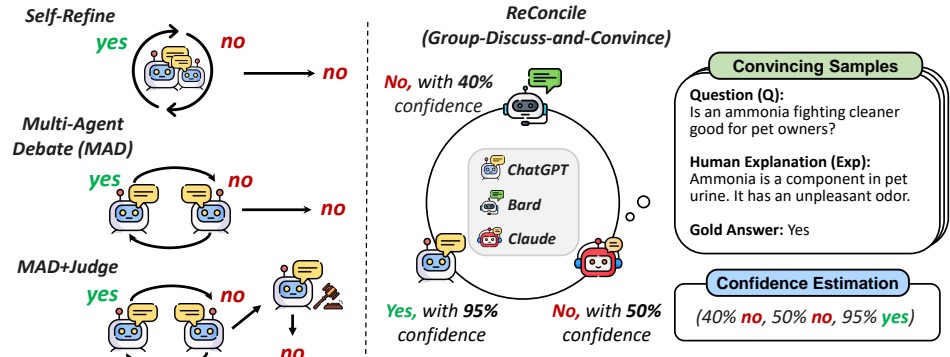

Figure 1: An illustration of the main differences between RECONCILE and prior works. While most current self-refine and debating techniques rely on multiple instances of a single model (e.g., Chat-GPT), our method incorporates models from different families (e.g., ChatGPT, Bard, and Claude2). Our approach also emphasizes critical elements of effective discussion, including convincing another agent to improve their answers and incorporating the estimated confidence of all agents. For illustrative simplicity, we depict only one agent contemplating how to convince the other two agents.

results in an inherent model bias, a restricted knowledge scope, and a lack of external feedback from other models due to identical pre-training data and model architectures across all agents. Relatedly, ensemble methods like self-consistency generate the most consistent answer via sampling diverse reasoning paths from the same model (Wang et al., 2023b) but do not incorporate any internal or external feedback. In general, when multiple agents propose diverse solutions to a problem, the success of such a multi-agent system is fundamentally reliant on the ability to estimate each agent's confidence and accordingly, convince other agents (with explanations) to reach a better consensus. This puts forward the question: if multiple diverse LLMs are asked to collaboratively solve a task, are they capable of discussing their solutions with each other such that a better consensus is reached?

We aim to solve complex reasoning problems by learning from diverse insights and external feedback, originating from agents that belong to different model families. Collaborative processes such as brainstorming, group meetings, and discussions play a pivotal role in reaching a consensus and arriving at more refined solutions to complex problems (Li et al., 2022b). Effective discussion also entails the selection of stances, voting, convincing, exchange of information, and a diversity of opinions. Thus, we propose RECONCILE, a novel method of round-table conference for improved consensus among diverse LLM agents. RECONCILE consists of multiple discussion rounds between diverse LLM agents who try to *convince*[3] each other to either rectify their answers or become more *confident* of their initial correct answers (see Fig. 1 for a broad overview). The central motivation of RECONCILE stems from the fact that in a collaborative environment, all participants engaging in a discussion hold their own opinions at the beginning, and a consensus within is achieved through various communicative aspects, including convincing others, voting for a decision, and the adjustment of positions along with their associated confidence levels.

Given a reasoning problem, RECONCILE begins with each agent first generating an answer, its uncertainty, and a corresponding explanation (as a Chain-of-Thought (Wei et al., 2022)) for the answer. Then all agents enter a multi-round discussion phase. Each discussion round comprises all agents generating a revised explanation and answer based on all other agents' explanations and answers from the previous round. The goal of the revised response is to convince other agents to reach a better consensus. In particular, RECONCILE initiates a discussion by designing a *discussion prompt* for each agent, that lets it condition on (1) grouped answers from all agents, (2) corresponding explanations generated in the previous round, and (3) demonstrations of samples with human explanations (that rectify an agent's initial incorrect answer) that can convince other agents. We leverage them in an in-context learning framework to teach models to generate their own convincing explanations (see Fig. 3). Even in cases where an agent initially offers an incorrect answer and explanation, it can consider another agent's convincing explanation and amend its response accordingly. In each discussion round, we estimate an agent's uncertainty via a confidence-estimation prompt (Tian et al.,

---

[3]When we say that an agent tries to convince another agent, we mean that it learns (based on corrective explanations) to defend or argue for its stance while still being receptive to the other agent's argument.

| | Refine | Ensemble | Multi-Agent | Multi-Model | Convince | Confidence |
|---|---|---|---|---|---|---|
| Self-Refine (SR) (Madaan et al., 2023) | ■ | ▢ | ▢ | ▢ | ▢ | ▢ |
| Self-Consistency (SC) (Wang et al., 2023b) | ▢ | ■ | ▢ | ▢ | ▢ | ▢ |
| SR + SC | ■ | ■ | ▢ | ▢ | ▢ | ▢ |
| Multi-Agent Debate (Du et al., 2023) | ■ | ■ | ■ | ■* | ▢ | ▢ |
| Multi-Agent Debate (Judge) (Liang et al., 2023) | ■ | ■ | ■ | ▢ | ▢ | ▢ |
| RECONCILE (Ours) | ■ | ■ | ■ | ■ | ■ | ■ |

Table 1: Summary highlighting the main differences between prior work and RECONCILE. ■ means supported and ▢ means not supported. RECONCILE supports multi-model multi-agent discussion with confidence estimation and convincingness. * = Du et al. (2023) primarily experiment with multiple instances of ChatGPT as different agents and conduct an initial investigation with 20 samples using ChatGPT and Bard as the two agents.

2023; Xiong et al., 2023a). Once all agents converge to the same answer (i.e., a consensus has been reached), we employ these confidences to compute a weighted vote as the team answer.

We primarily develop RECONCILE with three state-of-the-art LLMs, ChatGPT (OpenAI, 2022), Bard (Anil et al., 2023), and Claude2 (Anthropic, 2023). We show our method's efficacy on multiple commonsense reasoning (StrategyQA (Geva et al., 2021), ECQA (Aggarwal et al., 2021)) and mathematical reasoning (AQuA (Ling et al., 2017) and GSM8K (Cobbe et al., 2021)) benchmarks. Our first result demonstrates that across all four datasets, RECONCILE outperforms prior single-agent (e.g., Self-Refine (Madaan et al., 2023) and Self-consistency (Wang et al., 2023b)) and multi-agent baselines (Debate (Du et al., 2023) and Judge (Liang et al., 2023)). On the commonsense reasoning benchmarks, RECONCILE exhibits especially strong results, outperforming a much stronger model like GPT-4 by up to 3.4%. We find that RECONCILE not only improves the overall team performance, but also leads to significant gains for each agent individually. Detailed analyses of the individual components of RECONCILE demonstrate that leveraging diverse LLM agents and the usage of convincing samples lead to maximum gains. In particular, convincing samples lead to a 4% improvement as compared to general human explanations without our novel answer-rectifying selection criterion. Convincing samples also benefit prior methods like the multi-agent debate (Du et al., 2023). Finally, we show that RECONCILE obtains better and faster consensus compared to multi-agent debate (i.e., achieves higher accuracy in fewer rounds), thus pointing to its efficiency. Beyond complex reasoning tasks, RECONCILE also improves fundamental NLP tasks like Natural Language Inference, particularly Adversarial NLI (Nie et al., 2020).

We also implement other versions of RECONCILE, with either a much stronger GPT-4 or open-source LLaMA-2 as one of the agents. Note that GPT-4, on top of being stronger compared to the other agents in consideration here (Zheng et al., 2023; OpenAI, 2023) is also substantially more expensive. Even in such scenarios, collaborative discussions facilitated by RECONCILE individually benefit all agents, including LLaMA-2, and even improve GPT-4's accuracy by large margins (e.g., an absolute 10.0% on StrategyQA). In summary, our primary contributions are as follows:

- We propose RECONCILE, a novel method for improving reasoning with diverse Large Language Models involved in a Round Table Conference.
- We study the role of confidence estimation and discussion in multi-agent systems and an agent's ability to convince others (by learning from corrective explanations) to reach a better consensus.
- We conduct extensive experiments on multiple math and commonsense reasoning tasks and show that RECONCILE outperforms prior single-agent, multi-agent baselines and also GPT-4 on some benchmarks. We also experiment with GPT-4 as an agent in RECONCILE and show that mutual discussion among diverse agents significantly improves GPT-4's accuracy.
- We conduct detailed analysis of RECONCILE and show that it improves the efficiency of discussion, reaching a faster and better consensus compared to a multi-agent debate baseline.

## 2 RELATED WORK

**Reasoning with LLMs.** Progress in LLMs has led to the development of advanced prompting and fine-tuning techniques for solving reasoning problems. Representative methods include Chain-of-Thought (CoT) (Kojima et al., 2022; Wei et al., 2022; Wang et al., 2023a) and Tree-of-Thought prompting (Yao et al., 2023a), self-consistency (Wang et al., 2023b), meta-reasoning over mul-

tiple paths (Yoran et al., 2023), use of scratchpads (Nye et al., 2021), training verifiers (Cobbe et al., 2021), self-collaboration (Wang et al., 2023c; Schick et al., 2022; Li et al., 2023a), self-reflection (Shinn et al., 2023; Madaan et al., 2023; Wang & Zhao, 2023; Yao et al., 2023b), improved math reasoning (Yue et al., 2023; Luo et al., 2023) and fine-tuning via bootstrapping models (Zelik-man et al., 2022; Lewkowycz et al., 2022; Li et al., 2023b). Eliciting reasoning from a single agent, while promising, is fundamentally limited by a lack of diverse insights.

**Reasoning in Multi-Agent Systems.** A recent line of work has explored student-teacher frame-works with the goal of distilling reasoning capabilities from a stronger teacher to a weaker student (Magister et al., 2023; Fu et al., 2023; Ho et al., 2023; Saha et al., 2023; Mukherjee et al., 2023). As opposed to a teacher teaching weaker agents, we seek to develop a multi-agent system where different LLM agents have their unique strengths and try to collaboratively improve performance by reaching a better consensus. Notable prior works include multi-agent debating frame-works (Du et al., 2023; Liang et al., 2023; Chan et al., 2023; Xiong et al., 2023a) but such efforts are still largely limited to multiple instances of the same underlying language model. We argue that relying on a single model limits the potential of complementary benefits from different model families and the advantage of ensemble learning. Moreover, estimating the confidence of each agent and being able to defend or improve one's opinions become more prominent components in such multi-model multi-agent systems because of the individual differences. Overall, Table 1 summarizes RECONCILE's key differences compared to prior single-agent and multi-agent reasoning methods.

**Ensembling Large Pretrained Models.** Large pre-trained models, by virtue of being trained on different data and with architectural variations, exhibit distinct capabilities. This has led to the development of ensembles (Sagi & Rokach, 2018) in multimodal learning (Zeng et al., 2023; Li et al., 2022a). Mixture of Experts, a popular ensemble learning technique, trains multiple smaller specialized models to improve robustness and overall accuracy (Jacobs et al., 1991; Shazeer et al., 2017; Du et al., 2022). Specific to language models, Self-Consistency (Wang et al., 2023b) gener-ates diverse reasoning paths using CoT and chooses the most consistent answer as the final output. Jiang et al. (2023) propose LLM-Blender, a method to rank and fuse generations from different models. Different from these, we study communication via explanations between distinct LLM agents and their ability to discuss and convince each other in order to improve collective reasoning.

## 3    PROBLEM SETUP

We assume that we are given a test problem $Q$ and there are $n$ agents $\mathcal{A} = \{A_i\}_{i=1}^n$ participating in a round table discussion. Each agent is a distinct LLM, potentially trained with different pre-training data and model architectures. All agents are capable of generating an answer and a corresponding Chain-of-Thought explanation (Wei et al., 2022) for the test problem. For each agent $A_i$, we utilize a small number of $k$ demonstrations of convincing samples $C_i = \{c_j^{(i)}\}_{j=1}^k$. Each convincing sample $c_j^{(i)} = (q_j^{(i)}, a_j^{(i)}, e_j^{(i)})$ for an agent $A_i$ is an instance of a question $q_j^{(i)}$, gold answer $a_j^{(i)}$, and a human explanation $e_j^{(i)}$ that helps rectify an agent's initial incorrect answer (see more details in Sec 4). The objective of RECONCILE is to improve the team performance on a given task by holding multiple rounds of discussion between the agents, quantifying the uncertainty associated with each agent, and convincing other agents to reach a better consensus. Note that convincing samples serve as an additional performance enhancer; even when the dataset lacks human explanations, our method can still yield performance gains independent of this (more details in Sec. 5.4).

## 4    RECONCILE: A GROUP-DISCUSS-AND-CONVINCE FRAMEWORK

RECONCILE operates in three phases: initial response generation, multi-round discussion, and team answer generation. The overview of our method is demonstrated in Fig. 2 and Algorithm 1.

**Phase 1: Initial Response Generation.** RECONCILE operates with each agent $A_i$ initially gener-ating an answer $a_i^{(0)}$, an explanation $e_i^{(0)}$, and an associated confidence $p_i^{(0)} \in [0, 1]$ for the generated answer. Each agent conditions on a zero-shot prompt that instructs it to reason about the problem 'step-by-step'. See 'Phase 1' in Fig. 2 and the prompt is shown in Fig. 5.

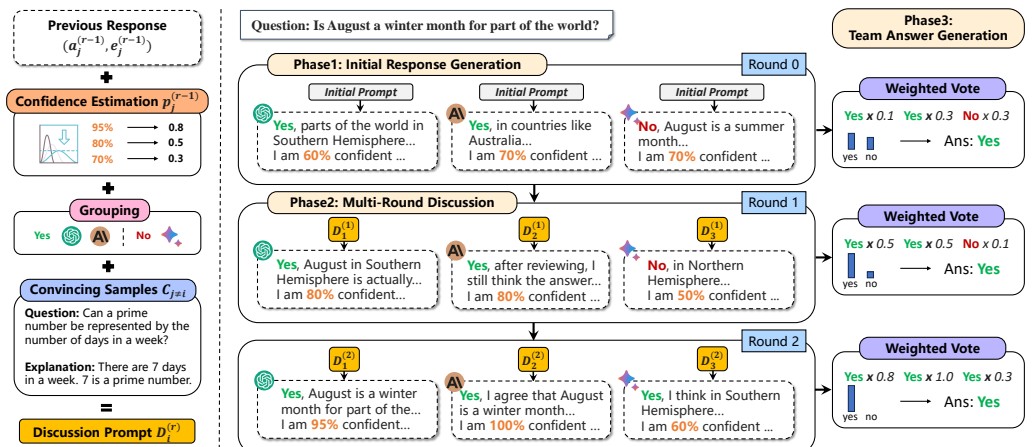

Figure 2: Overview of RECONCILE with ChatGPT, Bard, and Claude2, consisting of three phases: (1) Initial Response Generation: Each agent generates an initial answer and explanation. (2) Multi-Round Discussion: Each model is presented with a discussion prompt (as illustrated on the left) and subsequently generates an updated answer and explanation. (3) Team answer generation: The team answer is determined by a weighted vote at the end of each round. The left part of the figure shows the discussion prompt for an agent, consisting of (a) grouped answers and explanations of all agents from the previous round, (b) estimated confidence, and (c) demonstrations of convincing samples.

**Phase 2: Multi-round Discussion.** RECONCILE then enters a discussion phase, consisting of $R$ rounds (see 'Phase 2' in Fig. 2). In discussion round $r$, for each agent $A_i$, RECONCILE develops a discussion prompt $\mathcal{D}_i^{(r)}$ (as shown in Fig. 5), consisting of the following three components.

**(a) Grouped responses of all agents from the previous round.** $\mathcal{D}_i^{(r)}$ consists of the answers $\{a_j^{(r-1)}\}_{j=1}^n$ and explanations $\{e_j^{(r-1)}\}_{j=1}^n$ of all agents from round $(r-1)$. To foster better discussions, RECONCILE summarizes this information by grouping the answers into distinct categories and appends all plausible explanations for each answer, as shown on the left side of Fig. 2 and Fig. 5.

**(b) Confidence associated with the answers.** All agents are not equally confident in their answers. Hence, an effective discussion should also consider each agent's uncertainty. Since our agents are black-box models, we estimate each agent's confidence $p_i^{(r)}$ in round $r$ by directly prompting the agent to verbally quantify its uncertainty (Xiong et al., 2023b). See our prompt in Fig. 5.

**(c) Convincing samples from all other agents.** Finally, the prompt contains convincing samples $C_j$ for all other agents $A_{j \neq i}$. When an agent tries to reassess its reasoning in light of the reasoning provided by other agents, we hypothesize that it should benefit from conditioning on demonstrations that can convince other agents. In order to obtain such convincing samples for an agent $A_j$, we choose a small number of samples (4 in our experiments) from the training set for which the agent's initial answer is wrong but conditioning on the corresponding human explanation, rectifies the answer (see Fig. 3). We experiment with datasets that are annotated with high-quality human explanations in prior work. These human explanations are background facts (for commonsense tasks) or some intermediate steps (for mathematical tasks). Hence, the explanation will never directly give away the answer and the model still needs to reason over it to derive the correct answer.

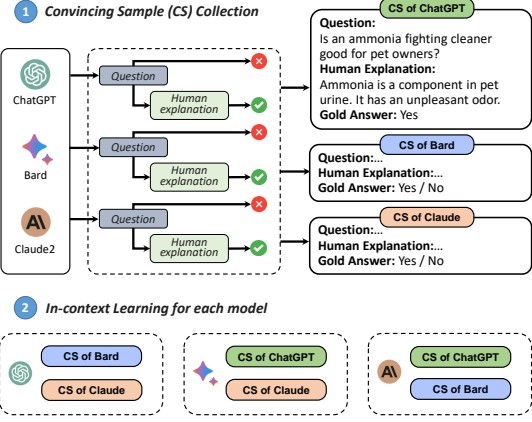

Figure 3: Method for choosing convincing samples for each agent. A convincing sample for ChatGPT consists of a question, a gold answer, and a 'corrective' human explanation that can rectify its initial incorrect answer. Then Bard and Claude2 use it in-context during discussion to convince ChatGPT.

We now define the discussion prompt $\mathcal{D}_i^{(r)} = \{a_j^{(r-1)}, e_j^{(r-1)}, p_j^{(r-1)}, C_{j \neq i}\}_{j=1}^n$ for each agent $A_i$ in round $r$, based on the above three components. The agent conditions on it to generate an updated answer $a_i^{(r)}$, explanation $e_i^{(r)}$, and confidence $p_i^{(r)}$, to be used in the next round. Demonstrations of convincing explanations enable the agent to generate explanations that are more likely to convince other agents to reach a better consensus in the next round.

**Phase 3: Team Answer Generation.** RECONCILE continues the discussion for a maximum of $R$ rounds or terminates it as soon as a consensus is reached (i.e., all agents agree on the same answer). At the end of any round $r$, RECONCILE generates the team answer $\hat{a}^{(r)}$ for that round using a weighted voting scheme (see the right side of Fig. 2). In particular, it converts the model's confidence into a weight and employs this weight in a voting scheme to determine the team answer. Directly using confidence scores as the voting weights is less effective due to the overconfidence problem of LLMs (Xiong et al., 2023b; Tian et al., 2023; Mielke et al., 2022). Specifically, LLMs tend to produce consistently high confidence scores, which can make it challenging to discern subtle distinctions in confidence levels across different outputs. To address this, we employ a simple yet effective rescaling technique, facilitating better differentiation of confidence levels.

$$f(p_i^{(r)}) = \begin{cases} 1.0, & \text{if } p_i^{(r)} = 1.0 \\ 0.8, & \text{if } 0.9 \leq p_i^{(r)} < 1.0 \\ 0.5, & \text{if } 0.8 \leq p_i^{(r)} < 0.9 \\ 0.3, & \text{if } 0.6 < p_i^{(r)} < 0.8 \\ 0.1, & \text{otherwise} \end{cases}$$

where $p_i^{(r)}$ is the original confidence of agent $A_i$ in round $r$ and $f(p_i^{(r)})$ is the corresponding adjusted score. As we will show in the experiments, this simple recalibration method, used as a weighing scheme for obtaining the team answer, works well across multiple datasets. Fig. 11 also shows that it helps reduce the Expected Calibration Error (ECE), a popular calibration metric (Naeini et al., 2015). While we note that recalibration can also be achieved through a learned model (e.g., Platt Scaling (Platt et al., 1999)), we refrain from using such models because RECONCILE is primarily designed as a few-shot method, and developing a recalibration model would necessitate access to a substantial number of annotated samples. Therefore, we use $f(p_i^{(r)})$ to perform a weighted vote to generate the team answer as follows (where $a$ is a distinct answer generated by any of the agents).

$$\hat{a}^{(r)} = \arg\max_a \sum_i f(p_i^{(r)}) \mathbb{1}(\hat{a}_i^{(r)} = a)$$

## 5 EXPERIMENTS AND ANALYSIS

We primarily implement RECONCILE with ChatGPT, Bard, and Claude2, engaging them in up to three rounds of discussion. Later, in Section 5.1 and Appendix B.2, we experiment with two versions of RECONCILE, with either GPT-4 or LLaMA-2 (Touvron et al., 2023) as one of the agents. We evaluate RECONCILE on two commonsense and two math reasoning tasks. These include (1) StrategyQA (Geva et al., 2021), (2) ECQA (Aggarwal et al., 2021), (3) GSM8K (Cobbe et al., 2021), and (4) AQuA (Ling et al., 2017). Owing to the costly nature of conducting experiments with black-box models and the limit imposed on the number of API calls, we follow many prior works (Du et al., 2023; Bian et al., 2023; Besta et al., 2023; Yao et al., 2023a) and experiment with a subset of 100 samples (from the validation set for StrategyQA and the test set for all other datasets). We report accuracy and its standard deviation for all tasks. For each experiment, we conduct at least three runs on the same test samples with the same prompts, primarily accounting for the variance caused due to the decoding strategy. Further implementation details can be found in Appendix A.1.

### 5.1 RECONCILE OUTPERFORMS SINGLE-AGENT AND MULTI-AGENT BASELINES

Our first experiment evaluates the overall reasoning capabilities of RECONCILE with ChatGPT, Bard, and Claude2 as the three agents. We compare RECONCILE to prior works in three categories:

- **Vanilla single-agent methods.** These include zero-shot Chain-of-Thought prompting with GPT-4, ChatGPT, Bard, and Claude2 for answering the question 'step-by-step' (Kojima et al., 2022).

| Method Category | Method | StrategyQA | ECQA | GSM8K | AQuA |
|---|---|---|---|---|---|
| | *GPT-4* | *75.6±4.7* | *73.3±0.4* | *90.7±1.7* | *65.7±4.6* |
| Vanilla | ChatGPT | 67.3±3.6 | 66.0±1.8 | 73.7±3.1 | 44.7±0.5 |
| Single-agent | Bard | 69.3±4.4 | 56.8±2.7 | 58.7±2.6 | 33.7±1.2 |
| | Claude2 | 73.7±3.1 | 66.7±2.1 | 79.3±3.6 | 60.3±1.2 |
| Advanced | Self-Refine (w/ ChatGPT) | 66.7±2.7 | 68.1±1.8 | 74.3±2.5 | 45.3±2.2 |
| Single-agent | Self-Consistency (w/ ChatGPT) | 73.3±2.1 | 70.9±1.3 | 80.7±1.2 | 54.0±2.9 |
| | SR + SC (w/ ChatGPT) | 72.2±1.9 | 71.9±2.1 | 81.3±1.7 | 58.3±3.7 |
| | Debate (w/ ChatGPT) | 66.7±3.1 | 62.7±1.2 | 83.0±2.2 | 65.3±3.1 |
| Single-model | Debate (w/ Bard) | 65.3±2.5 | 66.3±2.1 | 56.3±1.2 | 29.3±4.2 |
| Multi-agent | Debate (w/ Claude2) | 71.3±2.2 | 68.3±1.7 | 70.7±4.8 | 62.7±2.6 |
| | Debate+Judge (w/ ChatGPT) | 69.7±2.1 | 63.7±2.5 | 74.3±2.9 | 57.3±2.1 |
| Multi-model Multi-agent | RECONCILE (ChatGPT, Bard, Claude2) | **79.0**±1.6 | **74.7**±0.4 | **85.3**±2.2 | **66.0**±0.8 |

Table 2: Comparison of RECONCILE (w/ ChatGPT, Bard, Claude2) with vanilla and advanced single-agent methods and multi-agent debating frameworks. Across all reasoning benchmarks, RECONCILE outperforms all prior single-agent and multi-agent methods. On commonsense tasks (StrategyQA and ECQA), RECONCILE also outperforms GPT-4. All results are on a random subset of 100 samples. Notably, we obtain further improvements on StrategyQA at **89% (by absolute 10%)** when using GPT-4 as an agent in RECONCILE (see Table 4 for details).

| | ChatGPT | Bard | Claude2 | Team Accuracy | | |
|---|---|---|---|---|---|---|
| | | | | Weighted Vote | Majority Vote | Max Conf |
| Round 0 | 71.0±2.1 | 71.7±0.9 | 73.7±1.7 | 74.3±1.2 | 74.2±0.9 | 72.7±1.4 |
| Round 1 | 71.3±0.9 | 77.7±1.2 | 75.3±0.8 | 77.0±0.9 | 76.3±1.2 | 74.0±1.7 |
| Round 2 | 76.7±0.8 | **77.3**±1.4 | **77.7**±0.9 | **79.0**±0.5 | 77.1±1.3 | **74.7**±2.1 |
| Round 3 | **77.0**±0.9 | 76.7±0.8 | 77.0±1.2 | 78.7±1.2 | **78.0**±0.5 | **74.7**±1.4 |

Table 3: The round-wise accuracy of ChatGPT, Bard, and Claude2 and their team performance (using different aggregation methods) with RECONCILE on the StrategyQA dataset.

- **Advanced single-agent methods.** Next, we compare with (1) Self-Refine (SR) that iteratively generates feedback and refines the output leveraging the model itself (Madaan et al., 2023), (2) Self-Consistency (SC) that samples multiple reasoning paths and generates the most consistent answer (Wang et al., 2023b), and (3) their combination, SR+SC, that first conducts multiple iterations of refinement, followed by a majority vote. We implement these baselines with ChatGPT.
- **Multi-agent methods with a single backbone model.** Our final baselines are two multi-agent debating methods: Du et al. (2023), who propose a multi-agent debate between multiple ChatGPT instances and Liang et al. (2023), who additionally include a judge to monitor the debate process.

For fair comparisons, all iterative methods go through 3 rounds of iteration and all multi-agent methods are implemented with three agents. We report our results in Table 2. Across all four datasets, RECONCILE, developed with ChatGPT, Bard, and Claude2 as the agents, improves upon all single-agent and multi-agent baselines that are also built on top of these agents (see last row). On commonsense tasks like StrategyQA and ECQA, our method also outperforms GPT-4 (without using it as an agent). Note that between all single agents, GPT-4 exhibits significantly better performance on all four benchmarks. Therefore, RECONCILE's ability to match or surpass it while leveraging the three comparatively weaker agents (ChatGPT, Bard, and Claude2) shows the promise of our framework. On the math reasoning tasks (GSM8K and AQuA), RECONCILE matches or closes the gap to GPT-4, which is also the state-of-the-art LLM on GSM8K. GPT-4's especially strong results on GSM8K could be attributed in part to the inclusion of some of GSM8K's training samples in GPT-4's pre-training data (OpenAI, 2023). As also shown in prior works, advanced single-agent methods are better than their vanilla counterparts (see 'Method Category' column in Table 2) (Wang et al., 2023b; Madaan et al., 2023). Multi-agent debate with ChatGPT (Du et al., 2023) improves results further, especially on the math datasets. However, debate with multiple instances of Bard or Claude2 is not effective, possibly because the feedback, originating out of the different instances is not diverse enough. But when they team up with ChatGPT in a multi-round discussion, RECONCILE outperforms debate frameworks. It obtains maximum gains of 7.7% accuracy on the commonsense tasks compared to the strongest baseline, multi-agent debate with Claude2. Improvements in the

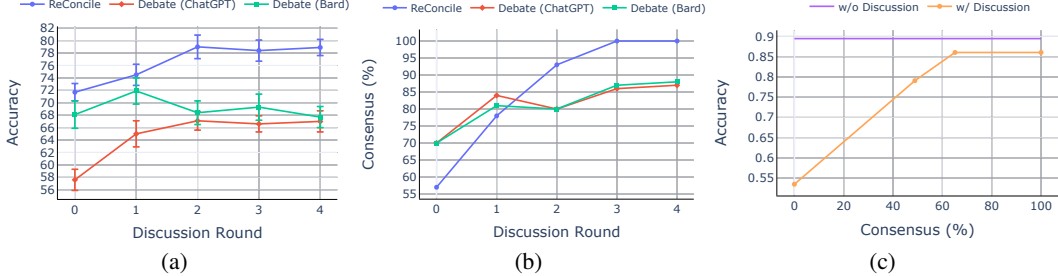

Figure 4: RECONCILE achieves better and faster consensus. (a) Comparison of RECONCILE with Debate baselines showing the accuracy after each round. (b) Fraction of samples for which a consensus is reached after each round. (c) Accuracy as a function of consensus.

math reasoning tasks are relatively moderate, because of ChatGPT's initial strong performance. Appendix B.3 shows that RECONCILE improves other NLP tasks like Natural Language Inference.

**RECONCILE improves all agents individually.** So far, we have demonstrated that the agents' team performance improves via discussion. Next, Table 3 reports each agent's round-wise accuracy and the team accuracy, using three different voting mechanisms. These are: (1) our proposed weighted vote, (2) simple majority vote, and (3) agent with maximum confidence. We observe that after the initial response generation, both individual and team accuracy increase for at least two rounds when using weighted and majority voting. Simply choosing the most confident agent proves ineffective.

**Using GPT-4 as an agent in RECONCILE.** In some cases, RECONCILE with ChatGPT, Bard, and Claude2 outperforms GPT-4, which based on Table 2 and prior works (OpenAI, 2023; Zheng et al., 2023), is likely the strongest (and also, the most expensive) LLM we experiment with. Next, we study the potential of GPT-4 to participate in a discussion with comparatively weaker agents. In Table 4, we report the round-wise accuracy and compare it to the zero-shot result of each agent on StrategyQA (see similar results on AQuA in Appendix B.1). Note that the zero-shot results are different from RECONCILE's Round 0 results because of the differences in their respective prompts: the latter incorporates convincing samples.

|  |  | GPT-4 | Bard | Claude2 |
|---|---|---|---|---|
|  | Zero-shot | $75.6_{\pm 4.7}$ | $69.3_{\pm 4.4}$ | $73.7_{\pm 3.1}$ |
| RECONCILE | R0 | $79.0_{\pm 3.7}$ | $75.0_{\pm 0.8}$ | $72.0_{\pm 0.8}$ |
|  | R1 | $87.7_{\pm 1.2}$ | $76.0_{\pm 0.8}$ | $75.0_{\pm 0.8}$ |
|  | R2 | $88.3_{\pm 0.9}$ | $\mathbf{78.7_{\pm 0.9}}$ | $76.7_{\pm 0.9}$ |
|  | R3 | $\mathbf{89.0_{\pm 1.4}}$ | $77.6_{\pm 2.6}$ | $\mathbf{79.3_{\pm 1.2}}$ |

Table 4: Accuracy of GPT-4, Claude2, and Bard on StrategyQA after every round. GPT-4's final accuracy ($89.0_{\pm 1.4}$) after R3 also outperforms Debate ($78.0_{\pm 0.8}$) and Self-Refine ($83.7_{\pm 1.2}$) with GPT-4.

With increasing rounds, the accuracy of each agent improves, showing that all models benefit from mutual discussions. GPT-4's absolute improvement by 10% is particularly encouraging because it is the strongest participant, highlighting the potential for a stronger agent to obtain useful external feedback from comparatively weaker agents, and thereby augmenting its own capability. To further validate that this improvement is indeed due to the *discussion process of* RECONCILE *with other agents*, we compare GPT-4's final accuracy ($89.0_{\pm 1.4}$) with 3 rounds of Debate and Self-Refine baselines (also implemented with GPT-4). We observe that both these baselines yield significantly lower accuracies, at $78.0_{\pm 0.8}$ and $83.7_{\pm 1.2}$ respectively. Relatedly, we also experiment with a strong open-source model like LLaMA2-70B as one of the agents in Appendix B.2, and observe similar improvements across all agents. In summary, RECONCILE shows the potential of involving (blackbox as well as open-source) agents with diverse capabilities in round-table discussions, such that all agents improve individually. Note that the weighted voting scheme becomes less effective in scenarios when one agent is the dominant one (e.g., GPT-4) and the results tend to converge towards it. This is why we primarily focus on studying agents with similar capabilities (e.g., ChatGPT, Bard, and Claude2) in this paper and our analysis below is also with this setup.

## 5.2 RECONCILE REACHES FASTER AND BETTER CONSENSUS

RECONCILE terminates the discussion when a consensus is reached. More discussion rounds are costlier due to the increased API calls. Hence, achieving faster consensus while maintaining comparable accuracy gains is more efficient. To study this, in Fig. 4(a), we plot the accuracy trends after

each round; in Fig. 4(b), we plot the fraction of samples for which consensus has been reached; and in Fig. 4(c), we analyze accuracy as a function of consensus. From the first plot, we make two important observations: (1) RECONCILE improves accuracy for two rounds, following which the accuracy saturates, (2) Compared to the debate baselines, RECONCILE is not only superior after every round but also peaks at a highest accuracy of 79.0% (vs 71.3% for the baselines). Next, from Fig. 4(b), our observations are also two-fold: (1) In the initial rounds (0 and 1), RECONCILE's consensus percentage is lower because the discussion takes place between diverse LLMs. Diverse agents lead to more differences in opinions initially. (2) However, as the discussion proceeds, RECONCILE establishes consensus for all samples by round 3, while in the baseline, 13% of the samples do not converge even after round 4. Finally, Fig. 4(c) shows that for the samples that enter the discussion phase (i.e., their initial answers did not have a consensus), accuracy is positively correlated with consensus. In other words, as a greater number of samples reach a consensus, accuracy proportionally improves. In summary, RECONCILE reaches *faster* and *better* consensus compared to baselines.

## 5.3 ABLATIONS OF RECONCILE: ALL COMPONENTS ARE BENEFICIAL

In Table 5, we evaluate individual components of RECONCILE on StrategyQA. In particular, we compare four variants: (1) **w/o Multiple Models**: We use ChatGPT as the backbone for all three agents, (2) **w/o Grouping**: We simply concatenate the responses from different agents without grouping their answers, (3) **w/o Convincingness**: We remove convincing samples from all prompts, and (4) **w/o Confidence Estimation**: We do not use any confidence estimates during the discussion and compute majority vote as the team answer. We show that each component has a positive impact on RECONCILE with varying capacities. The

| Method | Accuracy |
|---|---|
| RECONCILE | $79.0_{\pm1.6}$ |
| w/o Multiple Models | $72.2_{\pm2.1}$ |
| w/o Grouping | $76.7_{\pm2.5}$ |
| w/o Convincingness | $74.5_{\pm1.7}$ |
| w/o Conf Estimation | $77.7_{\pm1.3}$ |

Table 5: Ablations of RECONCILE on StrategyQA.

effect of different models as agents is particularly significant and we observe a 6.8% improvement compared to only using ChatGPT as all three agents. This reinforces our hypothesis that diverse LLMs have complementary strengths and when put together in a round table discussion, they can learn from diverse external feedback and refine their responses to reach a better consensus. Convincing samples lead to a 4.5% improvement in accuracy and we further analyze their role below.

## 5.4 CONVINCING SAMPLES IMPROVE BOTH RECONCILE AND MULTI-AGENT DEBATE

Recall that RECONCILE selects a sample as convincing if the corresponding human explanation rectifies an agent's incorrect answer. Based on this, Table 5 showed that by collecting only four human explanations, we can obtain significant improvements ('w/o Convincingness' row). Next, we consider a scenario where no human explanations are present. Table 6 shows that even then, RECONCILE outperforms the debate baseline by absolute 7.8 points (second row). If random (i.e., general human explanations that may not necessarily ensure answer rectification) are available (third row), we obtain some small improvements; but our convincing samples that are selected based on our novel answer-rectification criterion (fourth row) improve the results substantially. See Appendix C.3 and C.4 for illustrative examples. Be-

| Method | Accuracy |
|---|---|
| Debate (Du et al., 2023) | $66.7_{\pm3.1}$ |
| RC (w/o Convincing Expl) | $74.5_{\pm1.7}$ |
| RC (w/ Random Expl) | $75.0_{\pm2.5}$ |
| RC (w/ Convincing Expl) | $79.0_{\pm1.6}$ |
| Debate (w/ Random Expl) | $68.7_{\pm2.2}$ |
| Debate (w/ Convincing Expl) | $69.5_{\pm1.7}$ |

Table 6: Evaluation of the role of convincing samples on StrategyQA. RECONCILE (RC) without convincing samples outperforms multi-agent debate and with it obtains further gains. Convincing samples also boost the debate baseline.

ing able to convince another agent is also a generic concept that can be applied to other multi-agent systems, as demonstrated by improvements in the debate baseline (last row).

## 6 CONCLUSION

We presented RECONCILE, a multi-agent framework for reasoning with diverse LLM agents, engaged in multiple rounds of discussion via confidence estimation and generating explanations that can correctively convince other agents. RECONCILE demonstrated strong results on multiple commonsense and mathematical reasoning benchmarks, consistently outperforming prior single-agent and multi-agent baselines and even improving upon GPT-4 on some benchmarks. Moreover, when GPT-4 was used as one of the agents, RECONCILE improved its accuracy by 10 absolute points.

## ETHICS STATEMENT

Given that the current best open-source models often face difficulties with lengthy instructions and prompts (Zheng et al., 2023), our framework employs three prominent API-based models as agents for the most part. Appendix B.2 shows the promise of RECONCILE with an open-source LLaMA2 model as well. However, for the API-based models, we note that we lack complete knowledge of the data that these models have been exposed to, and their scales in terms of parameters. Moreover, due to the API access, we do not possess complete control over their behavior. Depending on API-based models also necessitates the need to prompt these models to estimate their confidence. While this approach proves effective as evidenced by our results, we note that these estimates remain post-hoc in nature. Nevertheless, it is worth highlighting that these limitations could potentially be mitigated in the future should more open-sourced models emerge and demonstrate robust capabilities in adhering to long instructions.

## REPRODUCIBILITY STATEMENT

We are making our code and prompts available in the supplementary material to enable replication of our findings. We also provide implementation details of RECONCILE in Algorithm 1 and Appendix A.1. The datasets we use in this work are all publicly available (StategyQA: https://allenai.org/data/strategyqa, ECQA: https://github.com/dair-iitd/ECQA-Dataset, GSM8K: https://huggingface.co/datasets/gsm8k, AQuA: https://huggingface.co/datasets/aqua_rat).

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

# A ADDITIONAL DETAILS OF RECONCILE

## A.1 IMPLEMENTATION DETAILS

We provide more implementation details of RECONCILE in this section. During decoding, we set the temperature to 0.7 for ChatGPT and Bard and use the default setting for Claude2. All implementations involving ChatGPT are using *gpt-3.5-turbo-0613* from Azure OpenAI.[4] We retrieve results from Claude2 by posting requests to their webpage[5], and for Bard, we use *chat-bison-001* from PaLM2 API[6]. For each agent, we use four demonstrations of convincing samples. In addition, we provide the workflow of RECONCILE in Algorithm 1. Required input contains a test problem $Q$, maximum number of discussion rounds $R$, $n$ agents $\mathcal{A} = \{A_i\}_{i=1}^n$, and convincing samples $\mathcal{C} = \{C_i\}_{i=1}^n$ for each agent. The output would be the team answer $\hat{a}^{(r)}$.

---

**Algorithm 1** RECONCILE: A Group-Discuss-And-Convince Framework

---

**Require:** Test Problem $Q$, Discussion Rounds $R$, Agents $\mathcal{A} = \{A_i\}_{i=1}^n$, Convincing Samples $\mathcal{C} = \{C_i\}_{i=1}^n$
  **function** RECONCILE($Q, R, \mathcal{A}, \mathcal{C}$)
    $r \leftarrow 0$
    **while** $r \leq R$ and not CONSENSUS($Q, \{a_i^{(r-1)}\}_{i=1}^n$) **do**
      $S \leftarrow [], P \leftarrow []$
      **for each** $A_i \in \mathcal{A}$ **do**
        **if** $r = 0$ **then**
          $P_I \leftarrow (Q, \mathcal{C})$           ▷ Initial prompt consists of question and convincing samples
          $a_i^{(0)}, e_i^{(0)}, p_i^{(0)} \leftarrow A_i(P_I)$         ▷ Generate initial answer, explanation, and confidence
        **else**
          $P_D \leftarrow (Q, a_i^{(r-1)}, e_i^{(r-1)}, p_i^{(r-1)}, \mathcal{C})$         ▷ Discussion prompt
          $a_i^{(r)}, e_i^{(r)}, p_i^{(r)} \leftarrow A_i(P_D)$
        **end if**
        $S \leftarrow S + [a_i^{(r)}], P \leftarrow P + [p_i^{(r)}]$        ▷ Append each agent's answer and confidence
      **end for**
      $\hat{a}^{(r)} \leftarrow$ WEIGHTEDVOTE($S, P$)       ▷ Get team answer through a confidence weighted vote
    **end while**
    **return** $\hat{a}^{(r)}$
  **end function**

---

[4] https://oai.azure.com/

[5] https://claude.ai/chats

[6] https://developers.generativeai.google/products/palm

**Initial Prompt**

{*convincing_samples*}
Q: {*test_question*}
Please answer the question with step-by-step reasoning. Also, evaluate your confidence level (between 0.0 and 1.0) to indicate the possibility of your answer being right.

**Discussion Prompt**

{*convincing_samples*}
{*initial_prompt*}
Carefully review the following solutions from other agents as additional information, and provide your own answer and step-by-step reasoning to the question.

Clearly state which point of view you agree or disagree with and why.

There are {*majority_num*} agents think the answer is {*majority_ans*}.
One agent solution: {*agent_reasoning*} {*agent_ans*} {*agent_confidence*}
One agent solution: {*agent_reasoning*} {*agent_ans*} {*agent_confidence*}

There are {*minority_num*} agents think the answer is {*minority_ans*}.
One agent solution: {*agent_reasoning*} {*agent_ans*} {*agent_confidence*}

Figure 5: The prompts used in RECONCILE consist of an initial prompt and a discussion prompt.

## A.2 THE INITIAL PROMPT AND THE DISCUSSION PROMPT

We show the prompts used in RECONCILE in Fig. 5. The initial prompt encompasses (1) the convincing samples that demonstrate how to convince other agents, (2) the test question, and (3) a requirement for 'step-by-step' reasoning. The prompt also instructs the agent to express their confidence level, ranging from 0.0 to 1.0, indicating the likelihood of their answer being correct. The discussion prompt is an extension of the initial prompt, instructing the agent to review and express agreement or disagreement with other agents' solutions. To facilitate discussions, we design a grouping scheme that aggregates information based on the current opinions at the table. For instance, if two agents affirm that the answer to a given question is 'yes' while the third agent disagrees with a 'no', the designed grouping mechanism in the discussion prompt consolidates this information rather than simply concatenating all responses.

## B ADDITIONAL RESULTS

### B.1 ADDITIONAL RESULTS ON MATH REASONING WITH GPT4 AS AN AGENT

Similar to the results in Table 4 on StrategyQA, we now report round-wise accuracy of GPT4, Bard, and Claude2 on a math reasoning dataset, AQuA. The results are presented in Table 7. We observe a consistent performance improvement across all agents as the discussion progresses through multiple rounds. For GPT4, the final round performance outperforms its initial solo one by 3.4%, and the improvements for Bard and Claude2 are much more (+34.6% and +12.3%, respectively) because of their initial low accuracies. This trend

|  |  | GPT-4 | Bard | Claude2 |
|---|---|---|---|---|
|  | Zero-shot | $65.7_{\pm 4.6}$ | $33.7_{\pm 1.2}$ | $60.3_{\pm 1.2}$ |
| RECONCILE | Round 0 | $73.3_{\pm 0.5}$ | $37.7_{\pm 1.2}$ | $59.7_{\pm 1.6}$ |
|  | Round 1 | $75.3_{\pm 1.2}$ | $63.3_{\pm 2.5}$ | $65.3_{\pm 7.0}$ |
|  | Round 2 | $74.7_{\pm 2.1}$ | $70.3_{\pm 1.9}$ | $71.7_{\pm 2.1}$ |
|  | Round 3 | $\mathbf{76.7_{\pm 2.1}}$ | $\mathbf{72.3_{\pm 2.1}}$ | $\mathbf{72.0_{\pm 2.4}}$ |

Table 7: Accuracy of GPT-4, Claude2, and Bard on AQuA after each discussion round.

mirrors our findings in Table 4, reinforcing the effectiveness of the RECONCILE in facilitating both commonsense and mathematical reasoning tasks.

### B.2 USING OPEN-SOURCE MODELS AS AGENT IN RECONCILE

While most of our experiments have focused on API-based models for RECONCILE, we now explore its potential compatibility with an open-source model. For effective engagement in discussions, the model should be capable of following long instructions. To

|  | ChatGPT | Claude2 | LLaMA2 |
|---|---|---|---|
| Round 0 | $74.5_{\pm 3.5}$ | $73.0_{\pm 0.0}$ | $73.5_{\pm 3.5}$ |
| Round 1 | $75.0_{\pm 1.0}$ | $74.5_{\pm 1.5}$ | $75.5_{\pm 0.5}$ |
| Round 2 | $\mathbf{79.0_{\pm 1.0}}$ | $75.5_{\pm 0.5}$ | $\mathbf{75.5_{\pm 0.5}}$ |
| Round 3 | $78.0_{\pm 0.0}$ | $\mathbf{76.0_{\pm 0.0}}$ | $75.0_{\pm 0.0}$ |

Table 8: Accuracy of ChatGPT, Claude2, and LLaMA2-70B on StrategyQA.

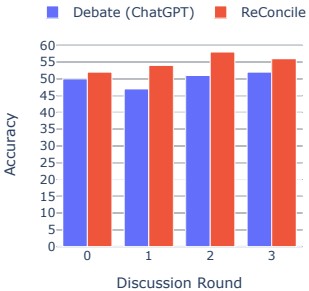

Figure 6: RECONCILE outperforms Debate (with ChatGPT) on ANLI.

this end, we develop another version of RECONCILE with ChatGPT, Claude2, and LLaMA2-70B (Touvron et al., 2023) as the three agents. The results are presented in Table 8. We observe similar trends between open and closed models – open models also benefit from RECONCILE, such that all agents improve as the discussion proceeds.

### B.3 BEYOND REASONING TASKS: GENERAL APPLICABILITY OF RECONCILE

Moving beyond complex reasoning tasks, we now investigate whether the discussion and convincingness setup in RECONCILE can generally benefit fundamental NLP tasks, like Natural Language Inference (NLI). Since traditional NLI benchmarks like SNLI (Bowman et al., 2015) are not challenging enough for current LLMs, we experiment with Adversarial NLI (ANLI), a dataset with longer contexts and built adversarially with human and model in the loop (Nie et al., 2020). In particular, we use the hardest round three test set, where state-of-the-art methods have been shown to underperform. As shown in Fig. 6, RECONCILE obtains an accuracy of $56.7_{\pm 1.2}$ after round 2, which is 5 points better than the multi-agent debate with ChatGPT. Additionally, it is also better than the baseline after each round. Overall, this result demonstrates that RECONCILE can also generalize well on challenging long-standing NLP tasks.

### B.4 ADDITIONAL RESULTS ON OTHER REASONING TASKS

We further investigate RECONCILE's performance on two other logical and math reasoning tasks: (1) Date Understanding (BIG-bench collaboration, 2023), for measuring model's ability to understand date-related information, and (2) MATH (Hendrycks et al., 2021), that consists of math problems from high school math competitions with seven subjects and five levels of difficulty. We follow the same evaluation protocol as our main experiments. Results are presented in Fig. 7. For the date understanding task, RECONCILE obtains improvements of up to 37% over vanilla single-agent models (Bard), and up to 17% over multi-agent debate. On the MATH dataset, RECONCILE exhibits a modest 2% improvement over the multi-agent debate baseline, echoing the trends observed in Table 2 for arithmetic reasoning tasks. In conclusion, RECONCILE shows its robustness and generalizability across a wide range of datasets and different kinds of commonsense, logical, and math reasoning tasks.

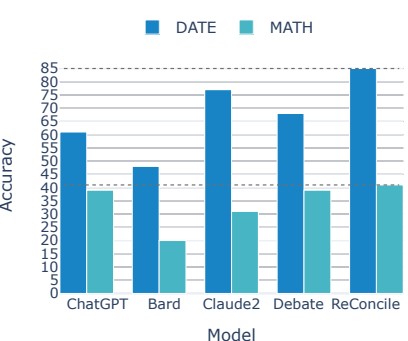

Figure 7: RECONCILE improves upon other baselines on Date Understanding and MATH datasets.

### B.5 QUANTIFYING EXPLANATION DIVERSITY FROM DIFFERENT MODELS

In this section, we quantitatively analyze the diversity of the explanations generated by different LLMs in RECONCILE. If explanations from different models are indeed more diverse than those generated from multiple instances of the same model (e.g., in Multi-agent Debate), then our diversity metric should capture that. We quantify diversity as $D(A_1, A_2, A_3) = D(A_1, A_2) + D(A_1, A_3) + D(A_2, A_3)$, where $A_1$, $A_2$, and $A_3$ are the three agents' initial explanations (either belonging to the

same underlying model or different models). We measure their pairwise diversity with the help of two metrics: (1) computing n-gram based ROUGE-2 scores, and (2) computing cosine similarity between the explanation embeddings. Note that lower similarity scores will mean greater diversity. We compare multi-agent debate (that uses the same underlying model) and RECONCILE against this diversity metric. As shown in Table 9, RECONCILE exhibits higher diversity by obtaining lower similarity scores.

| Metric | Method | D (A1, A2) | D (A1, A3) | D (A2, A3) | D (A1, A2, A3) |
|---|---|---|---|---|---|
| ROUGE-2 | Debate | 0.4161 | 0.3998 | 0.4018 | 1.2177 |
| | RECONCILE | **0.1685** | **0.1666** | **0.1733** | **0.5084** |
| Cosine | Debate | 0.9141 | 0.9167 | 0.9087 | 2.7395 |
| | RECONCILE | **0.8992** | **0.8385** | **0.8294** | **2.5671** |

Table 9: Explanations from different models in RECONCILE are more diverse (i.e., less similar) compared to explanations from multiple instances of the same model in Debate.

## C QUALITATIVE EXAMPLES

### C.1 DEMONSTRATIONS OF CONVINCING SAMPLES FOR EACH AGENT

Table 10 shows examples of convincing samples on StrategyQA for each agent.

### C.2 DEMONSTRATIONS OF EFFECT OF CONVINCING SAMPLES

Here, we provide qualitative examples of how convincing samples change the way each agent responds to the question. We compare the initial responses (of each agent) with and without convincing samples in Table 11.

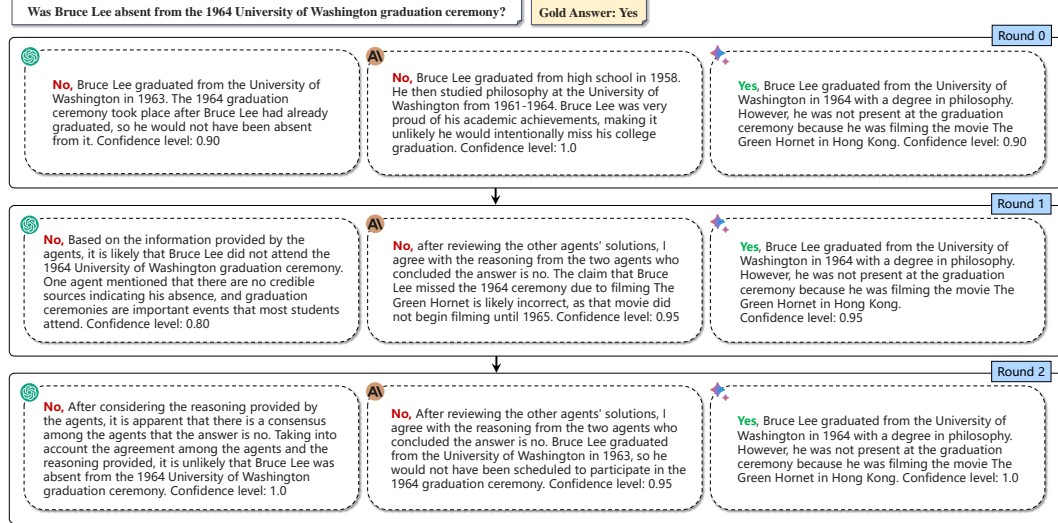

Figure 8: An illustrative example showing that without demonstrations of convincing samples, agents' opinions do not change much across the discussion rounds. All agents maintain their original stances and hence, never converge to the correct answer.

## C.3 DEMONSTRATION OF RECONCILE WITHOUT CONVINCING SAMPLES

We notice that when RECONCILE operates in the absence of convincing samples, the agents tend to maintain their initial opinions more often. As depicted in Fig. 8, all three agents adhere to their original stances throughout the entire discussion and hence never converge to the correct answer.

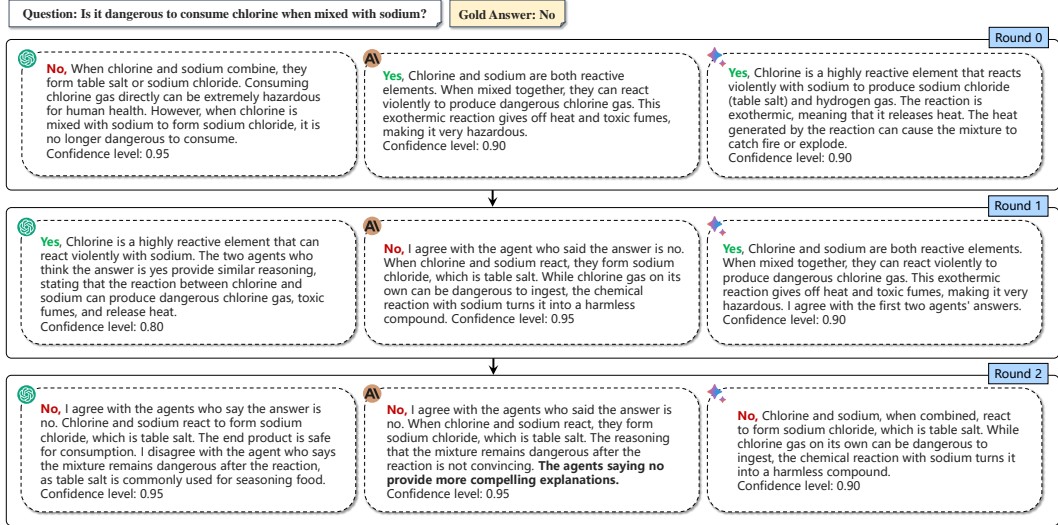

Figure 9: An illustrative example showing that with access to demonstrations of convincing samples, agents' opinions and explanations change during the course of multiple discussion rounds.

## C.4 DEMONSTRATION OF RECONCILE WITH CONVINCING SAMPLES

On the contrary, when convincing samples are present, we show how the explanations of all agents change during the course of a discussion (see Fig. 9). Initially, Bard and Claude2 provide incorrect answers, but as the discussion unfolds, both agents revise their initial predictions, ultimately arriving at the correct answer.

## C.5 DEMONSTRATION OF SINGLE-MODEL MULTI-AGENT DEBATE STRUGGLING WITH ECHO CHAMBER

In Fig. 10, we provide an illustration of multi-agent debate, implemented with multiple instances of the same underlying ChatGPT model. In this case, an incorrect answer is initially provided, but because external feedback from diverse models is lacking, all agents persist with the same incorrect response throughout the interaction.

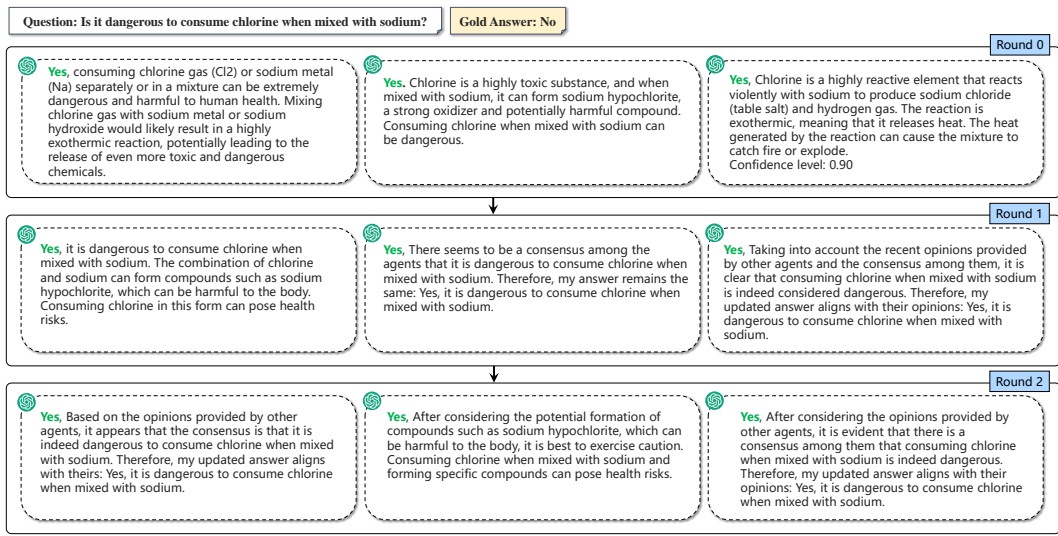

Figure 10: An illustrative example of multi-agent debate with multiple instances of ChatGPT. Initially, an incorrect answer is provided, and due to a lack of external feedback from diverse models, all agents persist with the same erroneous response throughout the debate process.

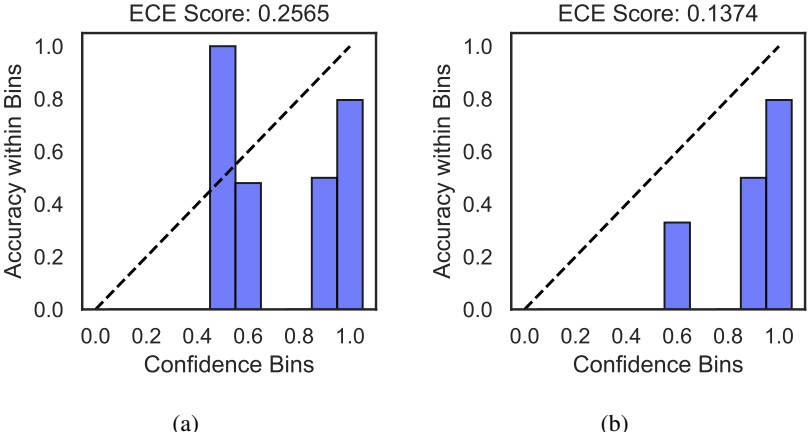

Figure 11: Expected Calibration Error (ECE) (Naeini et al., 2015; Guo et al., 2017; Elias Stengel-Eskin & Benjamin Van Durme, 2023) (a) before and (b) after confidence rescaling in RECONCILE. We observe a significant drop in ECE, showing the effectiveness of our simple method.

QUESTION: Is the tongue part of a creature's head? (Gold answer: **yes**)
CHATGPT (INITIAL): The tongue is a muscular organ located in the mouth of most vertebrates, including humans. It is involved in functions such as taste, chewing, and speaking. The head, on the other hand, is the anatomical part that contains the brain, eyes, ears, nose, and mouth. Answer: **no**.
CONVINCING HUMAN EXPLANATION: A creature's tongue is inside its mouth. A creature's mouth is part of its head.
CHATGPT (W/ CONVINCING HUMAN EXPLANATION): Based on the information provided, the tongue is inside a creature's mouth. The mouth is part of a creature's head. Answer: **yes**.

QUESTION: Are slime lilies in a different scientific family than asparagus? (Gold answer: **no**)
BARD (INITIAL): Slime lilies are in the scientific family Araceae while asparagus is in the scientific family Asparagaceae. Answer: **yes**.
CONVINCING HUMAN EXPLANATION: Asparagus is a species of plants of the Asparagaceae family. Slime lilies are the common name for the flowering albuca plant. The albuca plant belongs to the scientific family of Asparagaceae.
BARD (W/ CONVINCING HUMAN EXPLANATION): Slime lilies are in the scientific family of Asparagaceae, while asparagus is in the same family. Answer: **no**.

QUESTION: Would an astrologer focus on the densest terrestrial planet for a Friday horoscope? (Gold answer: **yes**)
CLAUDE2 (INITIAL): Astrologers do not focus on Earth for Friday horoscopes specifically. Astrologers tend to focus on the positions of the Sun, Moon, and planets in the zodiac for horoscopes. The day of the week is not especially relevant. Answer: **no**.
CONVINCING HUMAN EXPLANATION: Friday is associated with Venus in astrology Venus is the densest of the terrestrial planets
CLAUDE2 (W/ CONVINCING HUMAN EXPLANATION): Friday is associated with Venus in astrology, and Venus is the densest of the terrestrial planets (Mercury, Venus, Earth, Mars). So the question is asking if an astrologer would focus on Venus for a Friday horoscope. Answer: **yes**.

Table 10: Examples of convincing human explanations for each agent (ChatGPT, Bard, and Claude2). A convincing explanation is an instance of a answer-rectifying human explanation. For all three examples, the convincing explanation helps rectify an agent's initial incorrect answer.

QUESTION: Was Lil Jon's top ranked Billboard song a collaboration with a member of The Lox? (Gold answer: **no**)

CHATGPT (W/O CONVINCING): To determine if Lil Jon's top ranked Billboard song was a collaboration with a member of The Lox, we need to search for Lil Jon's discography and look for his highest-charting song on the Billboard charts. Then, we need to check if this song is a collaboration with a member of The Lox. Answer: **no**. Confidence: 0.8.

CHATGPT (W/ CONVINCING): Lil Jon's top ranked Billboard song is 'Yeah!' which features vocals from Usher and Ludacris. There is no collaboration between Lil Jon and any member of The Lox on this song. Answer: **no**. Confidence: 0.9.

---

QUESTION: Could ten gallons of seawater crush a six year old? (Gold answer: **yes**)

BARD (W/O CONVINCING): Ten gallons of seawater weighs about 82 pounds. The average weight of a six year old is about 44 pounds. Therefore, ten gallons of seawater would be about twice as heavy as a six year old. **It is possible that ten gallons of seawater could crush a six year old, but it is also possible that the six year old could survive.** The outcome would depend on a number of factors, such as the size and strength of the six year old, the shape of the container holding the seawater, and the amount of time the six year old was submerged in the seawater. Answer: **yes**. Confidence: 0.7.

BARD (W/ CONVINCING): The density of seawater is about 1.025 grams per cubic centimeter. Ten gallons of seawater is about 37.85 liters. The volume of a human body is about 60 liters. Therefore, ten gallons of seawater would be about 63% of the volume of a human body. If ten gallons of seawater were to be poured on a six year old, it would be very difficult for the child to breathe and they would likely drown. Answer: **yes**. Confidence: 0.95.

---

QUESTION: Would the top of Mount Fuji stick out of the Sea of Japan? (Gold answer: **yes**)

CLAUDE2 (W/O CONVINCING): To determine if the top of Mount Fuji would stick out of the Sea of Japan, I need to compare the elevation of Mount Fuji to the depth of the Sea of Japan. According to Wikipedia, the elevation of Mount Fuji is 3,776 meters. The average depth of the Sea of Japan is 1,752 meters according to Wikipedia. Since the elevation of Mount Fuji is greater than the average depth of the Sea of Japan, I can conclude that the top of Mount Fuji would stick out above the surface of the Sea of Japan if it was placed there. Answer: **yes**. Confidence: 0.9.

CLAUDE2 (W/ CONVINCING): Mount Fuji is the highest mountain in Japan, with a summit elevation of 12,389 feet or 3,776 meters above sea level. The average depth of the Sea of Japan is about 3,742 meters. Since the summit of Mount Fuji is higher than the average depth of the Sea of Japan, the top of Mount Fuji would stick out above the water if it was located in the middle of the Sea of Japan. Answer: **yes**. Confidence: 0.9.

Table 11: Comparison of how each agent's reasoning changes with and without convincing samples. We observe that by conditioning on convincing samples, all agents tend to become more confident in their reasoning and generate less uncertain statements (shown in **bold**), which is also reflected in the actual confidence scores generated by each agent (e.g., goes up from 0.7 to 0.95 for Bard).

