# OpenReview forum: "ReConcile: Round-Table Conference Improves Reasoning via Consensus among Diverse LLMs"
_ICLR.cc/2024/Conference — Submitted to ICLR 2024_

### Official Review · Reviewer_saRD · 2023-10-31

**Soundness:** 2 fair
**Presentation:** 3 good
**Contribution:** 2 fair
**Rating:** 3
**Confidence:** 4

**Summary:**

This paper introduces a multi-model, multi-agent framework called ReConcile, which is inspired by roundtable conference discussions. In each round, every agent must produce an answer, an explanation, and the corresponding confidence level. Following multiple rounds of discussion, ReConcile generates a collective team answer. Experiments on various benchmarks (StrategyQA, ECQA, GSM8K, AQuA) demonstrate the effectiveness of ReConcile.

**Strengths:**

1.It is an interesting idea to extend previous work from multi-agent debate to roundtable discussions.
2.Experiments on various benchmarks show that the proposed method outperforms previous work.

**Weaknesses:**

1.The experiments are conducted on relatively simple reasoning tasks, which seem inconsistent with the claim of "solving complex reasoning problems" made in the introduction. Does ReConcile enhance performance on more complex reasoning tasks, such as logical reasoning, the MATH dataset, CommonsenseQA?
2.The proposed method consists of multi-model predictions. An ablation study (Table 5) shows that the performance significantly decreases without the multi-model approach. It is unclear whether the performance gain is caused by the multi-model ensemble. A comparison with an ensemble baseline should be considered.
3.There are many hyperparameters in ReConcile (number of discussion rounds, voting weight). It would be interesting to explore if these hyperparameters can be dynamically generated during the roundtable discussions and if they are robust across tasks.
4.It is unclear whether a weak model could negatively impact the performance.
5.There is a line of work improving the reasoning ability of LLMs [1].   It would be interesting to investigate whether ReConcile outperforms these methods both theoretically and empirically.

**Questions:**

1.How to calculate the performance of ChatGPT? The performance reported in Table 2 seems lower than previous work [2]

[1] WizardMath: Empowering Mathematical Reasoning for Large Language Models via Reinforced Evol-Instruct
[2] Making Large Language Models Better Reasoners with Step-Aware Verifier

---

> ### Author Response · Authors · 2023-11-18
> **Response to Reviewer saRD (Part 1)**
>
> We thank the reviewer for their comments and for appreciating our idea of round-table discussions and the concerned experiments. Please find the answers to your questions below.
>
> > **The experiments are conducted on relatively simple reasoning tasks.**
>
> We respectfully disagree with the reviewer here because the tasks considered in our study are not simple. StrategyQA, CommonsenseQA, GSM8k, and AQuA are some of the most commonly used benchmarks in almost all prior works on reasoning. That said, following your suggestions, we conducted experiments on a few other datasets to further demonstrate the capabilities of ReConcile.
>
> * **CommonsenseQA**: Please note that CommonsenseQA experiments were already part of our paper and were referred to as ECQA. This is because we leverage convincing human explanations in ReConcile, which come from the ECQA benchmark (short for Explanations for CommonsenseQA) [1]. ReConcile obtains a 6.4% improvement in ECQA, as per Table 2 in the original version of the paper.
> * **Logical Reasoning**: StrategyQA is listed as a logical reasoning task as per the Big-Bench categorization of tasks [5], for which we already have results in the paper. We also experimented with another logical reasoning task, namely the “Date Understanding” task [2]. See results in the common response.
> * **MATH**: Following your suggestion, we also tested ReConcile on MATH. See results in the common response.
>
>
> > **It is Unclear whether the performance gain is caused by the multi-model ensemble.**
>
> Please refer to our analysis and discussion in the common response.
>
> > **A comparison with an ensemble baseline should be considered.**
>
> Apart from the vanilla single-agent methods, all of our baselines in Table 2 are already indeed ensemble-based (either involving a single agent like self-consistency or involving multiple agents like the Multi-agent Debate). Moreover, even for ReConcile, we compare different mechanisms for obtaining the "Team Answer" in Table 3 (e.g., majority vote, weighted vote, maximum confidence), all of which should be seen as ensemble methods for aggregating the answers from different models. In summary, most of the pertinent baselines in our paper are ensemble-based.
>
> > **There are many hyperparameters in ReConcile (number of discussion rounds, voting weight).**
>
> This is **not** true because the number of discussion rounds is not a hyperparameter. As Algorithm 1 of our paper shows, the number of discussion rounds is dynamically determined based on when all agents reach a consensus. R is just the "maximum" number of rounds, which can be any arbitrarily large value.
>
> As for the voting weights, we use the exact same weights across all seven tasks in the paper and in this rebuttal. This is a testament to its robustness. Please refer to our common response for a more comprehensive discussion of our recalibration strategy and voting weights.
>
> In summary, ReConcile does not have many hyperparameters, and our choice of voting weights is also adequately backed up by extensive robustness and ECE-style experiments.

---

> > ### Author Response · Authors · 2023-11-18
> > **Response to Reviewer saRD (Part 2)**
> >
> > > **It would be interesting to explore if these hyperparameters can be dynamically generated during the roundtable discussions and if they are robust across tasks.**
> >
> > Thanks for the suggestion, and we already noted this in the last few lines of Section 4. While one can try to learn the voting weights, it requires some training samples and we do not do so because ReConcile is primarily a few-shot framework.
> >
> >
> > > **It is unclear whether a weak model could negatively impact the performance.**
> >
> > Note that we do have experimental results in the paper using agents of varying strengths. In particular, Table 3 shows the results with GPT4 as one of the agents. We'd like to point you to the paragraph "_Using GPT-4 as an agent in ReConcile._" in Section 5.1 for details (and Reviewer 5C9y also acknowledges this in their strengths). Similarly, in Appendix B.2 in our original version, we have experiments with LLaMA-2 as one of the agents. Our conclusion is that through the discussion process in ReConcile, all agents individually improve (e.g., see GPT-4 improvement by 10% in Table 4). If the three agents have similar strengths, the team performance improves beyond all individual results, but if one agent is significantly stronger than others, the team answer expectedly converges to the strongest agent.
> >
> > > **There is a line of work improving the reasoning ability of LLMs. It would be interesting to investigate whether ReConcile outperforms these methods both theoretically and empirically.**
> >
> > Thanks for the reference! We note that a direct comparison between WizardMath and ReConcile is not relevant in the context of our work because the former is a single model while the latter is a generic reasoning framework with multiple models. Being a multi-model framework, ReConcile allows the flexibility of adding any model as an agent, including WizardMath.
> >
> > > **How to calculate the performance of ChatGPT? The performance reported in Table 2 seems lower than previous work.**
> >
> > We compute ChatGPT’s performance with zero-shot Chain-of-Thought prompting (see Sec 5.1 first bullet). Thanks for the reference although it does not seem to report ChatGPT’s performance but does so for text-davinci-002, which is an older version of the GPT series. In general, the performance of API-based models can always vary a bit because of their non-deterministic nature and also because different papers experiment with varying numbers of samples due to the computation cost [3, 4, 5]. In our paper, we make sure that our experiments are fair by (1) following the experimental setup of past work (Multi-agent Debate [3]), (2) reporting variance in accuracy for at least three separate runs, and (3) using a similar configuration (same rounds and same number of agents) of ChatGPT in baselines and ReConcile for fair comparisons.
> >
> >
> > [1] Explanations for CommonsenseQA: New Dataset and Models
> >
> > [2] https://github.com/google/BIG-bench/tree/main/bigbench/benchmark_tasks/date_understanding
> >
> > [3] Improving Factuality and Reasoning in Language Models through Multiagent Debate
> >
> > [4] Is ChatGPT a General-Purpose Natural Language Processing Task Solver?
> >
> > [5] A Systematic Study and Comprehensive Evaluation of ChatGPT on Benchmark Datasets

---

> > > ### Comment · Reviewer_saRD · 2023-11-20
> > >
> > > a weak model could negatively impact the performance
> > >
> > > -I am not asking if a strong agent could dominate/improve, but that ***a weak one*** or even several ones would severely harm the debating, including the final performance and also the number of rounds to reach a consensus.
> > >
> > > How to calculate the performance of ChatGPT? The performance reported in Table 2 seems lower than previous work.
> > >
> > > -I truly understand that you follow the most similar literature, which is reasonable. But I still suggest the authors follow some literature [1,2]  that shows much higher performance on Claude2/GPT4 with only CoT prompting, as the numbers seem not to ``vary a bit’’. I think this is very important to justify your performance gain (which is also achieved on prompting but more expensive).
> > >
> > > [1] MAMMOTH: BUILDING MATH GENERALIST MODELS THROUGH HYBRID INSTRUCTION TUNING
> > > [2] WizardMath: Empowering Mathematical Reasoning for Large Language Models via Reinforced Evol-Instruct

---

> > > > ### Author Response · Authors · 2023-11-21
> > > > **Response to additional comments from Reviewer saRD (Part 1)**
> > > >
> > > > Thank you for taking the time to read through our rebuttal and for the additional comments!
> > > >
> > > > > **I am not saying that the original datasets used are simple, but relatively simple as the major point of your motivation is to solve complex reasoning tasks.**
> > > >
> > > > The use of the adjective "complex" follows from prior works, which also typically refer to these reasoning tasks as "complex" (e.g., check the title or the first line of the abstract of all these papers [1,2,3]). That said, we are open to rephrasing/removing the word "complex" from our introduction in the final version.
> > > >
> > > > > **But unfortunately, as shown by your additional experiments here, the improvement on MATH is indeed very marginal**
> > > >
> > > > Yes, the improvement on MATH isn’t large (however it does get a moderate 2\% improvement, and achieves comparable results to GPT4 without relying on GPT4, using only a few-shot approach without any additional in-domain fine-tuning) and as you mentioned, it’s likely because the dataset itself is quite challenging. However, we should note that the focus of ReConcile is not to develop specific in-domain models for math reasoning. Instead, it is to present a general reasoning framework involving multiple diverse models that can discuss and correctively convince each other to reach a better consensus. In that regard, our significant improvements on challenging commonsense, logical, and NLI tasks like StrategyQA, Date Understanding, ANLI, underscore the effectiveness of ReConcile.
> > > >
> > > > > **Why not set this number as you stated "which can be any arbitrarily large value"?**
> > > >
> > > > Yes, we can set the number of discussion rounds to an arbitrary large value. In particular, please refer to Figure 4(a) in our paper, where we report results up to round 4. The performance of all methods saturate after round 2 and ReConcile continues to outperform the baselines after each round.
> > > >
> > > > > **Is it true that many prompts can reach a consensus within 3 rounds?**
> > > >
> > > > Yes, please refer to Figure 4(b) for this where we plot the consensus percentage after every round. The consensus percentage saturates after round 3. While ReConcile achieves 100% consensus, for the baselines, around 13\% of samples do not reach a consensus. In summary, combining the findings from Figure 4(a) and 4(b), we conclude that ReConcile reaches **faster** and **better** consensus (see Sec 5.2 for details).
> > > >
> > > > > **There should be a guideline about how to set these parameters.**
> > > >
> > > > Yes, the guideline is to recalibrate and use the corresponding weights (as described in Section 4 Phase 3). The recalibration scale that we use in the paper is robust and works well across all seven datasets.

---

> > > > ### Author Response · Authors · 2023-11-21
> > > > **Response to additional comments from Reviewer saRD (Part 2)**
> > > >
> > > > > **I am not asking if a strong agent could dominate/improve, but that a weak one or even several ones would severely harm the debating, including the final performance and also the number of rounds to reach a consensus.**
> > > >
> > > > To clarify further, we consider two scenarios in our paper with respect to stronger/weaker agents: (a) two stronger models (ChatGPT+Claude2) combined with one weaker model (LLaMA2) in Table 8, and (b) one stronger model (GPT4) combined with two weaker models (Bard+Claude2) in Table 4. In both these scenarios, we observe that all agents (stronger as well as weaker) improve individually through the discussion process. However, if one agent is stronger than the others, the final team answer will intuitively converge to the strongest agent.
> > > >
> > > > Now, it should be noted that in our studies, while some models are weaker than others, they are still strong enough to follow long complex instructions and carry out discussions. If some agents are not even capable of that, then their contribution to the discussion will be minimal. For example, if one agent’s initial accuracy is extremely low, then it is conceivable that it will have very little contribution in terms of complementary knowledge to the discussion process.
> > > > This is similar to what the Self-Refine paper [4] also reports on page 7 under “Does SELF-REFINE work with weaker models?” that Vicuna-13B is capable of generating initial outputs, but struggles significantly with the refinement process.
> > > >
> > > > > **I still suggest the authors follow some literature [1,2] that shows much higher performance on Claude2/GPT4 with only CoT prompting, as the numbers seem not to vary a bit.**
> > > >
> > > > Thank you for the references! We’ve added these to our revised version. The MAmmoTH paper indicates GPT-4's performance on GSM8K is 92.0. In ReConcile, we report GPT-4's score as 90.7±1.7, suggesting a potential high of 92.4, which we think is comparable. We again want to emphasize that API-based model performances can vary (also shown in [5] that the reported numbers are different from WizzardMATH), and we are using multiple trials to capture this variance.
> > > >
> > > > > **I think this is very important to justify your performance gain**
> > > >
> > > > We believe that fair comparisons with all the baselines and a comprehensive ablation study that systematically evaluates the contribution of each component justify the performance gain of ReConcile. Please let us know if you’d like us to clarify this further.
> > > >
> > > > [1] Chain-of-Thought Prompting Elicits Reasoning in Large Language Models
> > > >
> > > > [2] Self-Consistency Improves Chain of Thought Reasoning in Language Models
> > > >
> > > > [3] Least-to-Most Prompting Enables Complex Reasoning in Large Language Models
> > > >
> > > > [4] Self-Refine: Iterative Refinement with Self-Feedback
> > > >
> > > > [5] Is ChatGPT a General-Purpose Natural Language Processing Task Solver?

---

> > > > ### Author Response · Authors · 2023-11-21
> > > > **Request to Reviewer saRD**
> > > >
> > > > We tried our best to address your concerns by conducting additional experiments on new datasets (as you suggested) and performing other analyses that further back up the claims made in the paper. If these seem satisfactory, we’d request you to help revisit your scores. Please also let us know if there are any follow-up questions!

---

> > ### Comment · Reviewer_saRD · 2023-11-20
> >
> > the tasks considered in our study are not simple.
> >
> > -I am not saying that the original datasets used are ***simple***, but ***relatively*** simple as the major point of your motivation is to solve ***complex*** reasoning tasks. I know these are popular benchmarks for many works demonstrating LLM’s reasoning abilities. For example, why I mentioned MATH over GSM8K on math reasoning is that the performance on GSM8K can be ***relatively*** easy (through using CoT prompting)  [1,2] to get  around 85 on Claude-2 and even ~97 on GPT4 (Code Interpreter) alone. Therefore, I am not convinced that the reasoning complexity of GSM8K (as well as other benchmarks here) is able to demonstrate the usage of such a complex multi-round round-table prompting strategy. However, for MATH, I do not see significant performance improvement over GPT4 (around 40) by far even using various in-domain data fine-tuning methods. But unfortunately, as shown by your additional experiments here, the improvement on MATH is indeed very marginal (from 0.39 to 0.41).
> >
> > hyperparameters setting: number of discussion rounds and the voting weights
> >
> > -As stated in Sec5.1, "all iterative methods go through 3 rounds of iteration".  So why not set this number as you stated "which can be any arbitrarily large value"? Is it true that many prompts can reach a consensus within 3 rounds?
> > Also, from the additional results, the voting weights indeed affect the performance, so there should be a guideline about how to set these parameters.

---

### Official Review · Reviewer_5C9y · 2023-11-03

**Soundness:** 3 good
**Presentation:** 4 excellent
**Contribution:** 3 good
**Rating:** 8
**Confidence:** 4

**Summary:**

The paper presents a novel algorithm (ReConcile) for leveraging multiple LLM calls (like self-refine) to improve results over single LLM calls. ReConcile is similar to Multi-Agent Debate, but with 3 additional innovations:  (1) using distinct model families (GPT, Bard, & Claude), (2) leveraging confidence estimates from the models, and (3) encouraging the models to attempt to convince each other. ReConcile is compared against several relevant baseline algorithms on four standard benchmarks (StrategyQA, ECQA, GSM8K, AQuA). ReConcile obtains clear improvements over baselines on most of these experiments.

**Strengths:**

The method is well motivated and clearly explained, which is a challenge for a framework of this complexity. The baselines and benchmarks are well chosen, and the experimental methodology appears sound.

Ablation studies, summarized in Table 5, quantify the contribution of each of the architectural components to accuracy. The lift is largest for the multi-model dimension, which is particularly interesting.

Interestingly, it is noted (perhaps unsurprisingly) that when a stronger model (such as GPT-4) is included in multi-model, multi-agent debate, it tends to dominate the decision, which causes the results to converge on the accuracy of the stronger model acting alone.

**Weaknesses:**

While GPT-4 itself is shown to benefit from ReConcile, GPT-4 is not included in most of the baseline algorithms, such as Multi-Agent Debate. So we are left to wonder whether GPT-4 would benefit as much or more from those other baseline.

While relative efficiency is discussed (in terms of number of rounds for a given accuracy), token counts are not discussed. This makes it hard for readers to determine how much of the benefit of ReConcile might simply be due to increased tokens (and cost).

There are clearly many more important evaluations that could be performed. Fortunately the code for the framework is provided for review, so I assume it will be made available as OSS later. This will allow the community to run many more evaluations with new generations of models.

**Questions:**

How often do the agents settle on a single answer? Maybe I missed this detail somewhere.

---

> ### Author Response · Authors · 2023-11-18
> **Response to Reviewer 5C9y**
>
> We thank the reviewer for acknowledging the novelty and effectiveness of ReConcile and finding value in our experimental design and ablations. Please find the answers to your questions below.
>
> > **GPT-4 is not included in most of the baselines, such as Multi-Agent Debate.**
>
> Please note that we do have the baseline numbers with GPT-4 as an agent. Specifically, we want to point you to the caption of Table 4, which shows that Multi-agent Debate with GPT-4 obtains an accuracy of 78.0% and Self-Refine with GPT-4 has an accuracy of 83.7%. Note that these are the most pertinent and strongest baselines. Compared to these, ReConcile (with GPT-4 as an agent) obtains an accuracy of 89%, a significant 5.3\% improvement over the best baseline.
>
> > **While relative efficiency is discussed (in terms of number of rounds for a given accuracy), token counts are not discussed.**
>
> This is a great point! Although we do not explicitly talk about token lengths, we have ablations in the paper that point to the fact that ReConcile’s improvements are not because of increased token counts.
>
> - First, in Table 6, we compare the performance of ReConcile with random explanations and convincing explanations. Adding random explanations also increases the token count but significantly underperforms adding convincing examples (by 4%).
>
> - Second, following your suggestion, we also counted the average token length for ReConcile and Multi-agent Debate (w/ ChatGPT). They are 533 and 840 respectively. So, the strongest multi-agent debate baseline has a longer token length than ReConcile.
>
> In summary, the two major sources of improvement in ReConcile (multi-model and convincing samples) should not be attributed to the increased token length because their respective baselines either have similar or longer token lengths.
>
> > **There are clearly many more important evaluations that could be performed. Fortunately the code for the framework is provided for review, so I assume it will be made available as OSS later.**
>
> Yes, we will open-source our code. We are also looking forward to future work building on top of our implementation.
>
> > **How often do the agents settle on a single answer?**
>
> Agents settling on a single answer is basically what we refer to as "reaching a consensus". As per Figure 4(b) in the original version, after Round 3 for StrategyQA, all samples reach a consensus. The plot compares the consensus percentage after each round for all methods.

---

> > ### Comment · Reviewer_5C9y · 2023-11-18
> > **Response to rebuttal**
> >
> > Thank you for the clarifications and additional information!

---

### Official Review · Reviewer_b1xo · 2023-11-07

**Soundness:** 4 excellent
**Presentation:** 4 excellent
**Contribution:** 4 excellent
**Rating:** 8
**Confidence:** 4

**Summary:**

This paper proposes a multi-agent framework to improve the reasoning capabilities of LLMs by leveraging multiple rounds of discussion, demonstrations of answer-rectifying human explanations used to convincing other agents, and a confidence-weighted voting mechanism. The discussion phase also includes uncertainties in predictions with recalibrated confidence values. The evaluations of the proposed approach with three diverse agents (Claude, GPT-3.5 turbo and Bard) show significant performance improvements in some of the reasoning benchmarks where majority of the gains come from diversity in agent outputs and the use of convincing samples.

**Strengths:**

- The problem statement of leveraging multiple agents' diversity to improve reasoning is very timely.
- The paper presents a clear concise definition of the proposed approach as well as ablations of the individual components of the proposed approach in providing gains for reasoning benchmarks. The detailed ablations provide useful insights for the reader in building atop the proposed work.
- The gains on the StrategyQA benchmark are quite impressive from diversity of different model responses.

**Weaknesses:**

- The recalibration scale used to determine uncertainties in predictions is not backed by experiments

**Questions:**

- It would be useful to provide individual ablations for demonstrations of answer-rectifying human explanations on each of the models and more qualitative examples of the same to disentangle its effect from diverse model responses.

---

> ### Author Response · Authors · 2023-11-18
> **Response to Reviewer b1xo**
>
> We thank the reviewer for recognizing the contributions of our work and acknowledging our experiments and ablations. Please find the responses to your questions below.
>
> > **The recalibration scale used to determine uncertainties in predictions is not backed by experiments**
>
> Thanks, please refer to our common response for a detailed discussion on this.
>
> > **It would be useful to provide individual ablations for demonstrations of answer-rectifying human explanations on each of the models.**
>
> Thanks for the suggestion! We did not do this experiment specifically because "convincing or answer-rectifying human explanations" are used to prompt models to generate explanations that can convince _other_ agents. Hence, when there is only a single model, the role of convincing samples of other agents is unclear. That said, we experimented with including ChatGPT's _own_ convincing samples for its own predictions and observe moderate improvement in accuracy (from 67.3 to 69.0).
>
> > **More qualitative examples of the same to disentangle its effect from diverse model responses**
>
> Thanks for the suggestion! Please check Table 11 in the updated version of our paper which shows some examples of generated model reasoning with and without convincing samples. Our analysis suggests that in the presence of convincing samples, agents tend to become more confident of their reasoning and generate explanations that indeed sound more "_convincing_" to other agents. For example, the model starts its reasoning with factual statements like "_Mount Fuji is the highest mountain in Japan_" and avoids phrases like "_It is possible that …_", which may suggest uncertainty.

---

### Official Review · Reviewer_KtVY · 2023-11-10

**Soundness:** 3 good
**Presentation:** 3 good
**Contribution:** 3 good
**Rating:** 5
**Confidence:** 5

**Summary:**

This paper proposes a RECONCILE framework which leverages different LLM agents to collectively generates answers to reasoning tasks. Specifically, in multiple rounds, LLM agents generate responses and corresponding confidence using specific prompts, discuss in groups and generate explanations that may convince other agents. Experiments conducted on ChatGPT, Bard, and Claude2 achieve better performance on QA (StrategyQA and ECQA) and math problems (GSM8K and AQuA), compared to single-agent baselines.

**Strengths:**

1. This paper proposes an interesting multi-agent multi-round collection method to improve model performance on math and commonsense reasoning tasks. This indicates some complementary information from different trained LLMs and suggest how different LLMs can be applied to solve challenging tasks.
2. Some ablation studies and experiments (such as the importance of using confidence estimation) are informative and motivating.

**Weaknesses:**

1. There are many moving pieces in the propose method, especially the convincing samples and team answer generation heuristics with confidence estimation. In particular, the method requires choosing samples "from the training set for which the agent's initial answer is wrong" brings a very strong bias compared to the baseline methods. Although there are ablation studies, it is still not very convincing what "complementary benefits" are provided across different LLMs, and how exactly the multi-agent method is fundamentally better than previous debate or self-consistency. This is more concerning because in Table 3, it seems that although Weighted Vote saturates with 3 rounds, majority vote results keep improving. Furthermore, the proposed confidence rescaling heuristics seem brittle and not clear the impact of this from tasks to tasks.
2. This paper is mostly evaluated on two major tasks, QA/reasoning (StrategyQA and ECQA) and math (GSM8K and AQuA). Previous reasoning papers (such as self-consistency) were evaluated on a much wider range of tasks to illustrate how generalizable the proposed approach is.

**Questions:**

1. Why is the margin over baselines much larger for StrategyQA and ECQA (while being almost the same on GSM8K and AQuA)?
2. Are the results in Table 2 and Table 3 consistent (on StrategyQA)? Why is the weighted vote score 79.0 in Table 2 and 78.7 in Table 3?
3. Can you clarify how grouping works? What are "distinct categories"?
4. From Table 4, why do you think GPT4 + Bard/Claude2 hurt the performance (of GPT-4 + GPT-4) significantly? i.e., why would the powerful model be dramatically impacted by smaller models?
5. In Section 5.1, how is 3 rounds of Debate with GPT-4 different from RECONCILE with GPT-4?

---

> ### Author Response · Authors · 2023-11-18
> **Response to Reviewer KtVY (Part 1)**
>
> We thank the reviewer for acknowledging the effectiveness of ReConcile and the concerned experiments and ablations. Please find the answers to your questions below.
>
> > **Choosing samples "from the training set for which the agent's initial answer is wrong" brings a very strong bias compared to the baseline methods.**
>
> We want to highlight a few results here that will show that this is not the case.
>
> - First, note that ReConcile has many components to it and "choosing convincing samples" is only one of them. Even without these samples, ReConcile outperforms multi-agent baselines. In particular, Table 2 shows that even without access to these convincing samples, ReConcile achieves an accuracy of 74.2% on StrategyQA, substantially higher than single ChatGPT (67.3%) or multi-agent Debate with ChatGPT (66.7%) results. This demonstrates the utility of ReConcile beyond the usage of convincing samples.
>
> - Second, the utility of convincing samples goes beyond the specifics of the ReConcile framework and hence the concept of answer-rectifying explanations, in isolation, should be seen as an important contribution of our paper. Specifically, Table 6 demonstrates that convincing samples provide gains to baselines like multi-agent debate as well, improving the accuracy from 66.7% to 69.5%.
>
> - Third, this argument of bias also holds for papers like Chain-of-Thought prompting, which curate human-written demonstrations of step-by-step reasoning but then compare to "no-explanation" baselines.
>
> - Fourth, even if one were to consider usage of convincing samples as bias against the baselines, we only leverage 4 convincing samples in our experiments and hence the cost associated with obtaining those is negligible.
>
> >  **Although there are ablation studies, it is still not very convincing what "complementary benefits" are provided across different LLMs.**
>
> Conceptually, employing agents belonging to diverse LLMs brings in diversity in their reasoning processes (and the corresponding answers). This, in turn, fosters a better discussion, because if all agents are mostly agreeing on the same answer, there wouldn’t be any discussion. Better discussion leads to better consensus and improved accuracy. Please see our common response for more analysis of this complementarity and diversity of multiple models.
>
> > **How exactly the multi-agent method is fundamentally better than previous debate or self-consistency.**
>
> Figure 1 and Table 1 clearly explain the fundamental differences between ReConcile and other baselines. As also explained in Section 4, ReConcile differs from multi-agent debate in three distinct aspects: (1) usage of convincing samples (2) discussion between agents belonging to diverse models, and (3) usage of confidence estimation to quantify model uncertainty. Compared to self-consistency, ReConcile is a discussion framework while self-consistency employs a majority vote between different solutions from the same underlying model. Hence there is no notion of communication of explanations or iteratively improving/refining them in self-consistency. The ablation study presented in Table 4 methodically evaluates all key components by isolating one at a time, and the results show a notable drop in performance when any component is removed.
>
> > **In Table 3, it seems that although Weighted Vote saturates with 3 rounds, majority vote results keep improving.**
>
> Majority vote does not keep improving. We just chose to show results for 3 rounds in the paper but the results below should clarify that both voting mechanisms saturate in 3-4 rounds.
>
> |            | StrategyQA    |             | GSM8K        |             |
> |------------|--------------|--------------|--------------|--------------|
> | Round      | Majority     | Weighted     | Majority     | Weighted     |
> | Round 0    | 74.2         | 74.3         | 79.3         | 79.6         |
> | Round 1    | 76.3         | 77.0         | 81.0         | 82.7         |
> | Round 2    | 77.1         | 79.0         | 82.7         | 85.0         |
> | Round 3    | 78.0         | 78.7         | 83.3         | 84.3         |
> | Round 4    | 77.3         | 78.7         | 83.0         | 84.0         |

---

> > ### Author Response · Authors · 2023-11-18
> > **Response to Reviewer KtVY (Part 2)**
> >
> > > **The proposed confidence rescaling heuristics seem brittle and not clear the impact of this from tasks to tasks.**
> >
> > Please refer to our common response for a detailed discussion on this.
> >
> > > **This paper is mostly evaluated on two major tasks, QA/reasoning (StrategyQA and ECQA) and math (GSM8K and AQuA).**
> >
> > Our paper also had experiments on ANLI in Appendix B.3 (due to space constraints) and now we have added results for two more datasets, including Date Understanding and MATH. Please check the common response for more details.
> >
> > > **Why is the margin over baselines much larger for StrategyQA and ECQA (while being almost the same on GSM8K and AQuA)?**
> >
> > We think that this is because in StrategyQA, all individual models are mostly equally performant. Hence, ReConcile is able to better exploit the complementary strengths of these models to obtain bigger gains. On the other hand, in GSM8k, ChatGPT significantly outperforms Bard. Hence, multi-agent debate with ChatGPT also works really well for GSM8k, making the effect size with ReConcile comparatively smaller.
> >
> > > **Are the results in Table 2 and Table 3 consistent (on StrategyQA)?**
> >
> > Yes, they are! In Table 2, we report the accuracy after round 2, which matches with Table 3.
> >
> > > **Can you clarify how grouping works? What are "distinct categories"?**
> >
> > The distinct categories are the different answers produced by the different agents. So, in StrategyQA, if two agents have predicted the answer as "yes" and one has predicted a "no", there would be two groups. This enables us to consider the explanations generated by the agents at the abstraction of each group. Hence, during the discussion, we aggregate all agents' answers and explanations as "_there are 2 agents that think the answer is yes (with explanation x and y), and 1 agent that thinks the answer is no (with explanation z)_".
> >
> >
> > > **From Table 4, why do you think GPT4 + Bard/Claude2 hurt the performance (of GPT-4 + GPT-4) significantly?**
> >
> > Note that the GPT4+Bard+Claude2 accuracy for ReConcile is actually higher (89.0%) than the multi-agent debate baseline using GPT4 (78.0% accuracy, which has three GPT4 instances). This shows that GPT4 benefits from the diversity of the reasoning generated by somewhat less powerful models like Bard and Claude2.
> >
> > > **How is 3 rounds of Debate with GPT-4 different from ReConcile with GPT-4?**
> >
> > ReConcile with GPT4 is different from Debate with GPT-4 in the following three aspects: (1) it uses GPT-4, Bard, and Calude2 as the three agents (while Debate uses three instances of GPT-4 as three agents), (2) it makes use of "convincing samples" (there is no such thing in Debate), and (3) it uses uncertainty estimation for answers (again there is no such thing in Debate).

---

> > > ### Author Response · Authors · 2023-11-21
> > > **A Follow-Up on Our Responses**
> > >
> > > Dear reviewer, thanks again for your comprehensive review!
> > >
> > > Since we are nearing the end of the discussion period tomorrow (Nov 22), we wanted to politely follow up to see if our responses answered your questions satisfactorily (and we would greatly appreciate it if you could help revisit your scores accordingly). We are also happy to answer any other questions before the deadline. Thank you again!

---

> > ### Comment · Reviewer_KtVY · 2023-11-22
> >
> > Thanks for the responses.
> >
> > I would like to point out that choosing samples "from the training set for which the agent's initial answer is wrong" is fundamentally different from bias in CoT, where annotating wrong examples as a fix would fix some of the errors and directly boost model performance in the same distribution. Given the benefits shown in CoT and various few-shot prompting work, 4 is not "negligible".
> >
> > Regarding the number of rounds, can you provide more information with more rounds? Why do you think different methods saturates at different rounds?
> >
> > Can you answer my comment above about the whether confidence rescaling is generalizable to different tasks?

---

> > > ### Author Response · Authors · 2023-11-22
> > > **Response to follow-up from Reviewer KtVY**
> > >
> > > > **choosing samples "from the training set for which the agent's initial answer is wrong" is fundamentally different from bias in CoT, where annotating wrong examples as a fix would fix some of the errors and directly boost model performance in the same distribution. Given the benefits shown in CoT and various few-shot prompting work, 4 is not "negligible".**
> > >
> > > Thanks, as we mentioned in the previous response, even without using these convincing samples, ReConcile still outperforms the leading multi-agent debate baseline, which is also shown in Table 6 (74.5\% compared to 66.7\%). Moreover, the most relevant comparisons for using "convincing explanations" are with "no explanations" and "random explanations". We do this in Table 6, showing that convincing explanations outperform both these baselines. Lastly, Table 6 also demonstrates that convincing samples provide gains to baselines like multi-agent debate as well. If you believe there are other baselines against which we should test our "convincing explanations" to ensure an unbiased approach in selecting samples, we welcome your suggestions and are ready to conduct such experiments.
> > >
> > >
> > > > **Regarding the number of rounds, can you provide more information with more rounds? Why do you think different methods saturates at different rounds?**
> > >
> > > As we mentioned in the previous response, please note that ReConcile has a well-defined stopping criterion, which is when a consensus is reached. So, for StrategyQA, where all samples reach a consensus after 3 rounds (see Figure 4(b) in our original paper), our method terminates. Hence conducting more rounds will not alter the results (except for the small variance). We already added experiments for 1-2 more rounds just to empirically show that accuracy indeed saturates (and happy to add a few more rounds in the final version).
> > >
> > > > **Can you answer my comment above about the whether confidence rescaling is generalizable to different tasks?**
> > >
> > > Yes, as we mentioned in the previous response, it does generalize to all seven tasks that we experimented with in our paper. We’d like to point you to the entire comment "3. Robustness of ReConcile to our Recalibration Scale" in the general response that we wrote just to specifically address this.
> > >
> > >
> > >
> > > Overall, we have tried our best to answer your questions by conducting additional experiments on new datasets (as you suggested), and **we kindly request you to consider revising your scores (since today Nov 22 is the last day of discussion)**

---

### Author Response · Authors · 2023-11-18
**General Response (Part 1)**

We thank the reviewers for their detailed comments and for appreciating the ReConcile framework, our experimental design, and ablations. Based on the reviewers’ suggestions, we revised our paper with some new results and analyses. These changes are annotated in blue in the revised version. Below we provide a summary of these changes.

> **1. Results on new tasks**

While we already experimented with StrategyQA, CommonsenseQA, GSM8k, AQuA (and ANLI) that are some of the most commonly used benchmarks in almost all prior works on reasoning [1,2,3], reviewers saRD and KtVY were curious about the applicability of ReConcile on other logical and math reasoning tasks. Hence we experimented with the “Date Understanding” task [4] that the Chain-of-Thought paper also experimented with [1] and the MATH dataset [5]. We compare ReConcile with Multi-agent Debate (w/ ChatGPT) and single-agent baselines (as shown below). For Date Understanding, ReConcile obtains substantial improvements over any single model (up to 37%) and the leading multi-agent debate baseline (17%). For MATH, the improvements are moderate, about 2%. Please see Appendix B.4 and Figure 7 in our revised paper for more details.


|        | Claude2 | ChatGPT | Bard   | Debate (w/ ChatGPT) | ReConcile |
|--------|---------|---------|--------|---------------------|-----------|
| DATE   | 0.77    | 0.61    | 0.48   | 0.68                | **0.85**  |
| MATH   | 0.31    | 0.39    | 0.20   | 0.39                | **0.41**  |


Note that beyond challenging reasoning tasks, our paper also already experimented with ANLI [6], a challenging Adversarial Natural Language Inference task. Due to space constraints, we had to move these results to the appendix. Appendix B.3 in the original version reports a 5% improvement over the Multi-agent Debate (w/ ChatGPT) baseline, showing that ReConcile not only works well for reasoning tasks but also fundamental NLP tasks like NLI, as shown below.

|Round| 0  | 1  | 2  | 3  |
|----|----|----|----|----|
|Debate| 0.50 | 0.47 | 0.51 | **0.52** |
|ReConcile| 0.52 | 0.54 | **0.57** | 0.56 |

In summary, we show that our ReConcile framework generalizes and improves performance across a wide range of seven commonsense, logical, and math reasoning tasks (and other core NLP tasks like NLI).

> **2. Analysis of Diversity originating from multiple models in ReConcile**

Reviewers KtVY and saRD expressed interest in the diversity of the explanations generated by different models and how much that contributes to the improvements obtained by ReConcile. Here we provide two different analyses that indeed point to the fact that employing different models as agents in ReConcile brings greater diversity in discussion, leading to better accuracy.

- **(a) Ablation of ReConcile without Multiple Models.** First, the best way to measure the effect of diverse models is through an ablation study. We already did this in our original submission and the comparison between the first and second rows in Table 5 is direct empirical evidence of the gain obtained by the multi-model component of ReConcile. Reviewer 5C9y also acknowledges this in their strengths.

- **(b) Measuring Multi-Model Diversity with "Explanation Diversity Metric".** Our second analysis is one where we directly try to measure the diversity in the explanations generated by different models through an explanation diversity metric. If explanations from different models are indeed more diverse than those generated from multiple instances of the same model, then our diversity metric should be able to capture that. We quantify diversity between the explanations from three agents A1, A2, A3 (either belonging to the same underlying model or different models), as the summation of their pairwise diversities. We measure pairwise diversity with the help of two metrics (1) computing n-gram based ROUGE-2 scores, and (2) computing cosine similarity between the explanation embeddings. Note that lower similarity scores will mean greater diversity. We compare Multi-agent Debate (that uses the same underlying model) and ReConcile (that uses different models) against this diversity metric. As shown below, ReConcile exhibits higher diversity by obtaining lower similarity scores than Debate (please refer to Appendix B.5 and Table 9 in our revised version for details).


| Metric       | Method    | D(A1, A2) | D(A1, A3) | D(A2, A3) | D(A1, A2, A3) |
|--------------|-----------|-----------|-----------|-----------|---------------|
| ROUGE-2      | Debate    | 0.4161    | 0.3998         | 0.4018    | 1.2177        |
| ROUGE-2             | ReConcile | **0.1685**    |**0.1666**   | **0.1733**    | **0.5084**        |
| Cosine       | Debate    | 0.9141    | 0.9167    | 0.9087    | 2.7395        |
| Cosine   | ReConcile | **0.8992**    | **0.8385**    | **0.8294**    | **2.5671**        |

---

> ### Author Response · Authors · 2023-11-18
> **General Response (Part 2)**
>
> > **3. Robustness of ReConcile to our Recalibration Scale**
>
> Finally, reviewers KtVY, b1xo and saRD were also curious about the robustness of ReConcile to our recalibration scale. Below we provide three pieces of evidence supporting the robustness and effectiveness of our recalibration scale.
>
> - First, we use the exact same scale across all seven tasks (including the two new tasks experimented with in this rebuttal). This is a testament to its robustness.
> - Second, beyond the end task accuracy where we obtain improvements, the confidence recalibration also helps reduce the Expected Calibration Error (see Fig 11 in the Appendix of our updated version and Fig 10 of our original version) which further points to the usefulness of our calibration scale and the voting weights.
> - Third, we also manually experiment with a few different scales and show that the impact on the end task accuracy, as a result of it, is minimal. In the paper, we used scales of  [1.0, 0.8, 0.5, 0.3, 0.1]. We also tried other scales, as shown below
>
> w1: [1.0, 0.9, 0.7, 0.5, 0.3]
>
> w2: [1.0, 0.9, 0.5, 0.3, 0.1]
>
> w3: [1.0, 0.8, 0.6, 0.4, 0.2]
>
> w4: [1.0, 0.75, 0.5, 0.25, 0.0]
>
> Majority: Regardless of the confidence, simply choose the most-voted answer
>
>
> The results on StrategyQA and GSM8k, corresponding to each of the above scales, are as follows.
>
>
> | Voting weight | StrategyQA | GSM8K |
> |---------------|------------|-------|
> | w1            | 0.77       | 0.84  |
> | w2            | 0.79   | 0.83   |
> | w3            | 0.78       | 0.82   |
> | w4            | 0.77       | 0.83   |
> | Majority      | 0.76       | 0.83   |
> | Ours          | 0.79   | 0.85  |
>
>
> Combining all these findings, we conclude that our calibration strategy and confidence-weighted voting are simple, effective, and robust across seven different tasks.
>
>
> [1] Chain-of-Thought Prompting Elicits Reasoning in Large Language Models
>
> [2] Self-Consistency Improves Chain of Thought Reasoning in Language Models
>
> [3] Reasoning with Language Model Prompting: A Survey
>
> [4] https://github.com/google/BIG-bench/tree/main/bigbench/benchmark_tasks/date_understanding
>
> [5] Measuring Mathematical Problem Solving With the MATH Dataset
>
> [6] Adversarial NLI: A New Benchmark for Natural Language Understanding

---

### Meta-Review · Area_Chair_gL8F · 2023-12-05

**Metareview:**

The reviewers appreciate the nature and timeliness of proposed work and the fact that through "discussion" we can transfer information from one LLM to another. They also liked the ablations presented in the paper.

However they also bring up some concerns such as the fact that the work is only evaluated a few simple tasks (compared to related works), lack of clarity as to the contributions of each component, some results lacking experiments, and some claims made in the paper.

The authors provide comprehensive responses that are convincing and also conduct additional experiments to address the reviewers' concerns. Overall two reviewers are very supportive (8) and two are not (5, 3).

**Justification For Why Not Higher Score:**

While overall I believe this paper should be published, I recognize the concerns raised by KtVY and saRD. For saRD specifically I think their concerns are legitimate (I do not agree that they ignored the author's response).

**Justification For Why Not Lower Score:**

N/A

---

### Decision · Program_Chairs · 2024-01-16

Reject